# Mapping nucleosome-resolution chromatin organization and enhancer-promoter loops in plants using Micro-C-XL

Linhua Sun [1,2,5], Jingru Zhou[1,5], Xiao Xu[1,5], Yi Liu[2], Ni Ma[1,3], Yutong Liu[1], Wenchao Nie[4], Ling Zou[4], Xing Wang Deng [1,2] ✉ & Hang He [1,2] ✉

Although chromatin organizations in plants have been dissected at the scales of compartments and topologically associating domain (TAD)-like domains, there remains a gap in resolving fine-scale structures. Here, we use Micro-C-XL, a high-throughput chromosome conformation capture (Hi-C)-based technology that involves micrococcal nuclease (instead of restriction enzymes) and long cross-linkers, to dissect single nucleosome-resolution chromatin organization in *Arabidopsis*. Insulation analysis reveals more than 14,000 boundaries, which mostly include chromatin accessibility, epigenetic modifications, and transcription factors. Micro-C-XL reveals associations between RNA Pols and local chromatin organizations, suggesting that gene transcription substantially contributes to the establishment of local chromatin domains. By perturbing Pol II both genetically and chemically at the gene level, we confirm its function in regulating chromatin organization. Visible loops and stripes are assigned to super-enhancers and their targeted genes, thus providing direct insights for the identification and mechanistic analysis of distal CREs and their working modes in plants. We further investigate possible factors regulating these chromatin loops. Subsequently, we expand Micro-C-XL to soybean and rice. In summary, we use Micro-C-XL for analyses of plants, which reveal fine-scale chromatin organization and enhancer-promoter loops and provide insights regarding three-dimensional genomes in plants.

Three-dimensional (3D) genome organization is important for regulating gene expression, ensuring genome stability, and maintaining genome function. Central questions in 3D genomics include determining the mechanisms by which chromatin is folded and compressed within the nucleus, as well as the mechanisms by which 3D chromatin structures affect gene expression and cell fate. In 2009, first-generation high-throughput chromosome conformation capture (Hi−C) technology was established for unbiased identification of "all to all" interactions between any two regions throughout the genome. Subsequent optimization of this technology led to in situ Hi−C[1], Hi−C 2.0[2], and Hi−C 3.0[3], refining the resolution from the megabase scale to the kilobase scale. The resolution was substantially enhanced by Micro-C and Micro-C-XL through the inclusion of cleavage by micrococcal nuclease (MNase), rather than restriction endonuclease; this enabled the achievement of nucleosome resolution in yeast and mouse cell lines[4–7]. Various Hi−C and Micro-C variants, either target factors

[1]Peking University Institute of Advanced Agricultural Sciences, Shandong Laboratory of Advanced Agricultural Sciences at Weifang, Shandong 261000, China. [2]School of Advanced Agriculture Sciences and School of Life Sciences, State Key Laboratory of Protein and Plant Gene Research, Peking University, Beijing 100871, China. [3]PKU-Tsinghua-NIBS Graduate Program, Academy for Advanced Interdisciplinary Studies, Peking University, Beijing 100871, China. [4]Wuhan Frasergen Bioinformatics Co., Ltd., Wuhan 430075, China. [5]These authors contributed equally: Linhua Sun, Jingru Zhou, Xiao Xu. ✉e-mail: deng@pku.edu.cn; hehang@pku.edu.cn

guided by specific antibodies or loci captured by probes or open chromatin enriched technologies, were developed for the biased enhancement of local resolution[8–12]. These techniques provide powerful tools for high-resolution analyses of interactions between *cis*-regulatory elements (CREs) and genes.

In the past decade, the use of the popular restriction enzyme-based Hi−C and its derived C-technology revealed hierarchical 3D genome organization among prokaryotic and eukaryotic organisms[13,14]. Analyses in plants have revealed that, from larger scales (megabase-scale resolution) to finer scales (kilobases-scale resolution), chromosome territories, A/B chromatin compartments, topologically associated domains (TAD)-like structures, and chromatin loops can be identified by 3C, 4C, Hi−C, Capture-C, ChIA-PET, and HiChIP (reviewed in[14]). At larger scales, chromatin structure is more conserved; at smaller scales, greater structural diversity is present[15]. Furthermore, chromatin structures differ between plants and animals in terms of patterns, mechanisms, and functions. However, despite the use of 4-cutter enzymes such as *Mbo*I and *Dpn*II, the maximum resolution in *Arabidopsis* is ~1 kb[16,17]. Ultra-fine scale mapping of chromatin structures requires further exploration; the most widely used techniques involve restriction endonucleases, which are insufficient for analyses of chromatin structures at sub-kilobase resolution. The limitations of restriction endonucleases are most clearly demonstrated in analyses of the model plant *Arabidopsis thaliana*. The *Arabidopsis* genome is crowded and compact, containing more than 33,000 genes within a 135-Mb genome[18]. The mean lengths of gene bodies and intergenic regions are approximately 2–3 kb[16]; thus, the identification of chromatin structures between CREs (e.g., enhancers and genes) requires very high-resolution data.

Thus far, ≥4 types of common chromatin loops have been identified in animals, including architectural loops, enhancer–promoter loops, gene loops, and Polycomb-mediated loops[19]; however, the presence of these structures in plants remains unclear. Although various omics, genetics, and imaging techniques have been used for comprehensive analyses of the 3D genome of *Arabidopsis*, along with descriptive profiling of large-scale chromatin structure[16,20–22], the regulatory relationships of CREs and protein-coding genes (PCGs) have not been fully characterized. An understanding of the crosstalk between the one-dimensional (1D) genome and the 3D genome is necessary to fully characterize the structure and function of chromatin organization patterns in plants. Moreover, single-locus analyses in plants have shown that local chromatin structure contributes to transcriptional regulation[23]. Notably, 3D fluorescence in situ hybridization on root epidermal cells revealed that chromatin reorganization around the homeodomain transcription factor *GLABRA2* (*GL2*) locus is required to control position-dependent cell fate decisions[24]. A chromatin loop encompassing the *PINOID* (*PID*)/*AUXIN-REGULATED PROMOTER LOOP* (*APOLO*) intergenic region relies on APOLO-triggered H3K27me3 deposition around the *PID/APOLO* region[25]. In the early phase of vernalization, a gene loop involving physical contact between the upstream and downstream regions of the *Arabidopsis* floral repressor *FLOWERING LOCUS C* (*FLC*) locus is disrupted[26]. Photoreceptor phytochrome B (phyB) directly triggers the formation of a repressive H3K27me3-associated chromatin loop to facilitate the repression of *ARABIDOPSIS THALIANA HOMEOBOX PROTEIN 2* (*ATHB2*) in a light-dependent manner through interactions with VERNALIZATION INSENSITIVE 3-LIKE1/VERNALIZATION 5 (VIL1/VRN5), a component of PRC2[27]. Additionally, genome-wide identification of 10,044 putative enhancers (median size 347 bp) in *Arabidopsis* was conducted based on intergenic DNase I hypersensitive sites that are distant from any promoters (>1.5k upstream of transcription start sites [TSSs] and downstream of transcription termination sites [TTSs])[28]. Recently, super-enhancers (SEs) were identified as clustered non-promoter accessible chromatin regions, with sizes >1.5 kb (mean, 2.1 kb) and ≥5 accessible chromatin regions, via a comprehensive

analysis of DNase-Seq data from 17 diverse tissues/ecotypes[29]. Thus, in addition to the identification and characterization of enhancers in *Arabidopsis* and other species (e.g., rice, wheat, and maize) that became popular through 1D genomic approaches[28–33], there is an urgent need for a fine-scale comprehensive analysis method that can explore enhancer interactomes, thereby supporting efforts to identify relationships between chromatin organization and transcriptional regulation.

There are gaps and urgent demands to resolve the high-resolution chromatin interaction maps in plants, which is a technical bottleneck for dynamics, molecular mechanisms, and functional dissection studies. Therefore, in the present study, we modified Micro-C-XL and used it to explore fine-scale chromatin organization patterns in *Arabidopsis*. Micro-C-XL enables global analysis of chromatin organization compared with Hi−C, ranging from small-scale short-range chromatin loops to chromosome territories. The strong relationship between RNA Pols (e.g., Pol II) and local chromatin arrangements identified by Micro-C-XL indicates that RNA Pol II plays important roles in the formation of local chromatin domains. The genomes were divided into separate blocks based on compartment domains, which were populated by H3K27me3 or H3K9me2/H3K27me1. SEs were linked to observable stripes and loops, offering valuable insights for the identification and analysis of distal CREs in plants. In comparison with the traditional 1D methods for identifying enhancers in plants, such as ATAC-seq and DNase-seq, which can only predict enhancers through intergenic open chromatin regions, Micro-C-XL can clearly and unbiasedly observe the loops between enhancers and promoters. Potential regulators of these chromatin loops, such as *Microrchidia* (*MORC*) adenosine triphosphatase (ATPase) and RNA Pol V, were also identified. Micro-C-XL was then extended to crops such as rice and soybean, demonstrating its potential for use in the analysis of diverse plant species. Our findings will help to advance the overall understanding of chromatin organization and transcriptional regulation in plants.

## Results
### Micro-C-XL reveals fine-scale chromatin organization in *Arabidopsis*

To overcome the resolution limit (~1000 bp) in *Arabidopsis* 3D genome research, we used Micro-C-XL to map nucleosome-resolution chromatin organization. Micro-C-XL was conducted using MNase rather than restriction enzymes such as *Hind*III[21,22], *Dpn*II[16,20], or *Mbo*I[17] (Methods and Fig. 1a). Additionally, to yield more *cis* (intra-chromosomal) interactions and reduce the number of random ligation products, we combined formaldehyde (FA) and ethylene glycol bis (succinimidyl succinate) (EGS) to enhance local visibility based on previous approaches to optimize crosslinking chemistry[3] (Fig. 1a).

Sequencing analyses of each ~200-Gb library revealed >400 million valid contacts (Fig. 1b and Supplementary Data 1). The merging of two *Arabidopsis* Micro-C-XL libraries yielded 841 million valid contacts (Fig. 1b and Supplementary Data 1). On average, 1,412, 286, and 190 contacts per 200-bp analysis location, respectively, were achieved using our Micro-C-XL method, the in situ Hi−C method established in our previous study[17], and the Micro-C method (validated in mouse embryonic stem cells)[5]. Each Micro-C-XL library had a *cis*/total ratio of > 80% (Supplementary Data 1), indicating that the libraries were successfully constructed. Scaling plots of distance-dependent decay of chromatin interaction intensity generated by Micro-C-XL analysis in two biological replicates demonstrated good reproducibility (Supplementary Fig. 1a). Micro-C-XL rep1 and rep2 exhibited robust correlations that supported the similar contact patterns at 200-bp, 400-bp, and 2000-bp resolutions (Supplementary Fig. 1b–d). The stratum-adjusted correlation coefficient scores between biological replicates, determined by HiCRep, confirmed the biological reproducibility of Micro-C-XL (Fig. 1c). A scaling plot of chromatin contact

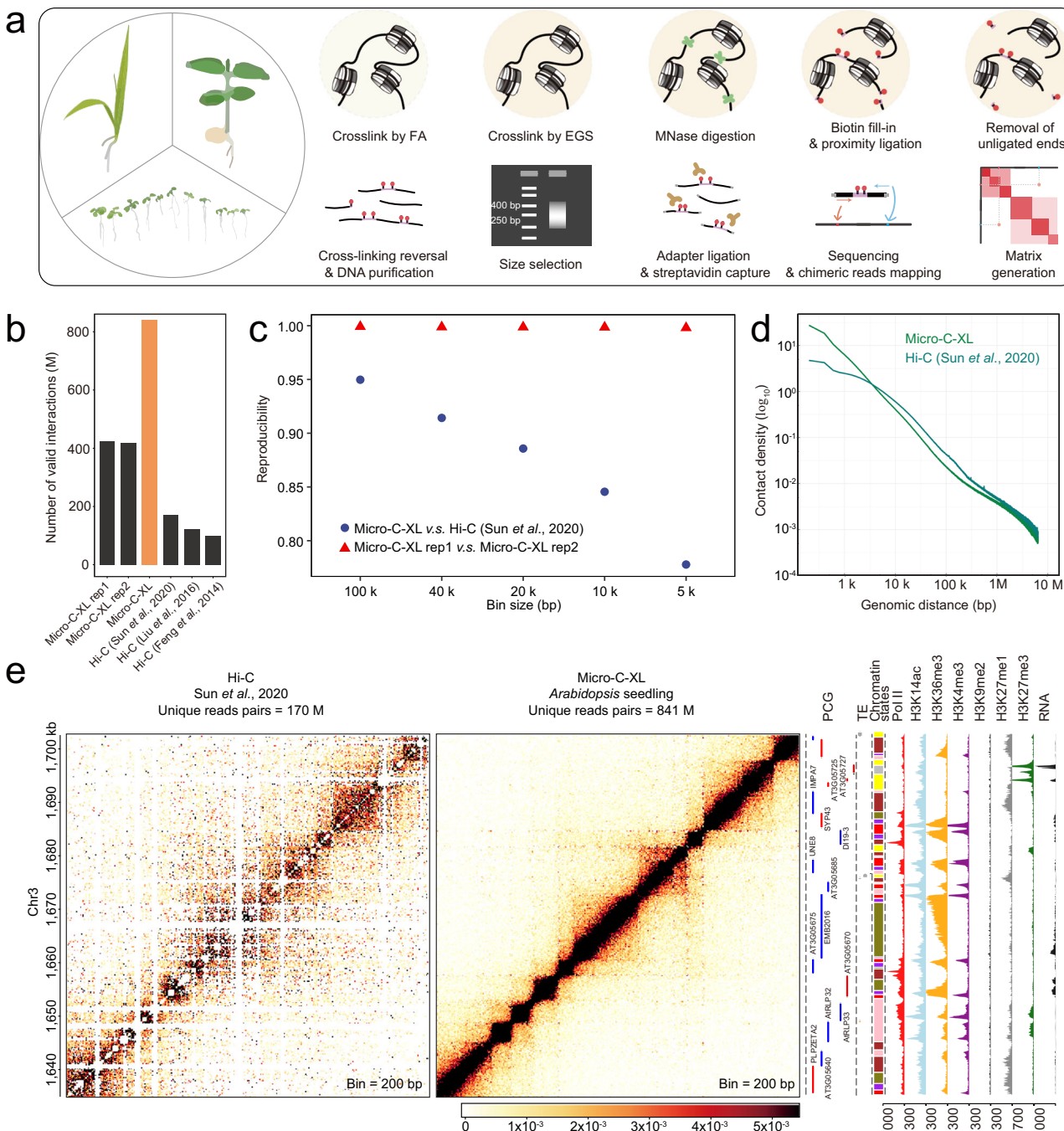

**Fig. 1 | Micro-C-XL mapping of nucleosome-resolution chromatin organization in *Arabidopsis*. a** Schematic of Micro-C protocol. Nuclei were extracted from plant tissue and then separately cross-linked by formaldehyde (FA) and ethylene glycol bis (succinimidyl succinate) (EGS). Subsequently, chromatin was digested with micrococcal nuclease (MNase), and DNA overhangs were filled with biotin-dNTP. After proximity ligation, unligated ends were removed by exonuclease. After cross-linking reversal and DNA purification, dinucleosomal DNA was selected for adapter ligation and biotin capture. Next, DNA was PCR-amplified and subjected to paired-end sequencing. **b** Bar plot showing the number of valid interactions in *Arabidopsis* identified by Micro-C, corresponding biological replicates, and Hi-C from three studies that used different restriction enzymes. **c** Stratum-adjusted correlation coefficient (SCC) bar plot showing good reproducibility of two Micro-C-XL biological replicates and Micro-C-XL vs. Hi-C[17]. **d** Scaling plot showing distance-dependent decay of contact density for Micro-C-XL and Hi-C[17], using 200-bp bins. **e** Random locus showing local chromatin organization of Hi-C[17] and Micro-C-XL using 200-bp bins along with multiple epigenomic datasets, Pol II occupancy, blocks of chromatin states, protein-coding genes (PCGs), and transposable element (TE) annotations. Color bars show contact intensities of Hi-C, and Micro-C-XL in the same ranges. *Arabidopsis* Hi-C map[17] was used as a control in subsequent figures.

density with increasing genomic distance indicated that Micro-C-XL (downsampled to an equal number of valid counts, compared with Hi-C) has stronger short-range chromatin contact density and slightly weaker long-range chromatin contact density, compared with Hi-C from multiple sources[16,17,20,21]; this comparison demonstrated the

advantages of Micro-C-XL in terms of studying *cis* local interactions (Fig. 1d and Supplementary Fig. 1e, f). Distinct *KNOT* structure, long-range chromatin loops, and other compartmental domains were clearly visible when using Micro-C-XL, implying that Micro-C-XL is comparable to Hi-C for the analysis of those large-scale structures

(Supplementary Fig. 2a–c). Since restriction endonuclease recognition sites are unevenly distributed in the genome, Micro-C-XL outperformed Hi–C in terms of chromatin structure, even at the same number of valid counts after subsampling (Supplementary Fig. 2d). Thus, Micro-C-XL outperformed Hi–C with respect to the clarity of chromatin organization patterns at nucleosome resolution (200-bp) (Fig. 1e). Many local chromatin organization patterns are shown in the Micro-C-XL contact map, along with various annotations (Fig. 1e and Supplementary Fig. 2d).

In summary, we successfully performed a Micro-C-XL analysis of *Arabidopsis*; it was clearly superior to the widely used Hi–C, particularly at the local scale.

## Chromatin domain boundaries in the *Arabidopsis* 3D genome
Various distinct local chromatin domains/boundaries are evident in the Micro-C-XL heatmap; however, the molecular mechanisms underlying the formation of these structures remain unknown (Figs. 1e and 2a). To identify and characterize these boundaries throughout the genome, we used the insulation score method commonly performed in animals. First, we comprehensively calculated the insulation score using Micro-C-XL interaction maps with different resolutions (from 200 bp to 5000 bp) at various scales (Fig. 2a). Next, by comparing insulation score distributions with the original interaction map, we revealed the insulation score at 200-bp resolution; the locations of corresponding boundaries appropriately segmented the 3D genome of *Arabidopsis* (Fig. 2a and Supplementary Fig. 3). By combining metagene and heatmap analyses using the insulation score distributions in various resolutions across boundaries identified using the 200-bp resolution map, we demonstrated that the *Arabidopsis* genome can be clearly segmented at the nucleosome scale, which is substantially better than segmentation at other larger scales (Fig. 2b). Subsequently, we validated the boundaries established at 200-bp resolution by stacking analysis using Micro-C-XL maps with different bin sizes, which indicated that most chromatin intensity reduction occurred at the boundaries (Fig. 2c). Using 200-bp resolution metrics, we identified 14,671 fine-scale chromatin domain boundaries in *Arabidopsis* (Supplementary Data 2).

Next, we investigated the genetic and epigenetic features at these chromatin boundaries. We found that the boundaries were closely associated with gene organization, typically adjacent to genes, and specifically enriched in promoters; their flanking regions had a median length of 5.6 kb (Fig. 2d, e). Chromatin state (CS) analysis indicated that boundaries with the greatest strength were mainly distributed in CS1 and CS2 (Fig. 2f, g). CS1 was mainly present in genome segments occupied by active histone modifications and variants, usually exhibited a low nucleosome density, and was associated with transcribed regions and TSSs. CS2 was similar to CS1, although it also contained high levels of H3K27me3[34]. Moreover, transcription factor enrichment analysis using chromatin boundaries revealed numerous transcription factors associated with gene transcription, including ELONGATED HYPOCOTYL 5 (HY5), NUCLEAR FACTOR Y, SUBUNIT B2 (NF-YB2), and AGAMOUS-Like 15 (AGL15) (Fig. 2h).

To further establish the relationship between chromatin boundaries and various epigenetic modifications or protein factors, as well as chromatin accessibility, we constructed a generalized linear model based on one-dimensional data such as chromatin strength from Micro-C-XL and a series of ChIP-Seq/ATAC-Seq (Supplementary Fig. 4 and Supplementary Data 3). Chromatin accessibility (ATAC-Seq) and the active histone mark H3K4me3 are positive predictors of boundary strength, while H3K9me2, H3K27me3, and H3K36me3 are negative predictors of boundary strength (Supplementary Fig. 4). The results predicted by the model align perfectly with the observed typical loci (Fig. 2a, Supplementary Figs. 3 and 4), indicating that the strength of the boundary is consistent with the distribution of chromatin accessibility. Many studies in both mice and humans have reported similar

close relationships between chromatin accessibility and chromatin boundaries[4,5,35]. It is noteworthy that another independent work recently focused on these self-interacting domains[36]. By applying Hi–C analysis to *Arabidopsis* Col-0 and using the same insulation score strategy (except at 1-kb resolution), 4794 interacting boundaries were identified and the domains between them were defined as compartment domains (CDs)[36]. High-level chromatin accessibility was observed on all chromosome arms, with a negative correlation between the insulation score and chromatin accessibility at the borders of CDs[36], which is entirely consistent with our observations of typical loci and results based on the generalized linear model. Using different technologies, our work and this independent work[36] have found and cross-verified that chromatin boundaries display high chromatin accessibility.

Overall, nucleosome-resolution insulation scores accurately analyzed the boundaries of heterogeneous chromatin domains; these boundaries were closely associated with active transcription and chromatin accessibility.

## Gene transcription influences local chromatin domain patterns in *Arabidopsis*
At 200-bp resolution, *Arabidopsis* Micro-C-XL maps exhibited extensive local chromatin domains occupied by ≥1 PCG (Figs. 1e and 2a), similar to the chromosomal interacting domains demarcated by promoters in yeast. *Arabidopsis* Hi–C maps identified these local chromatin domains, but the results were ambiguous.

RNA polymerase II (Pol II)-dependent transcription can shape local chromatin domains in yeast and mouse embryonic stem cells[5–7,37]. To investigate the relationships between gene transcription and chromatin organization in *Arabidopsis*, we integrated Micro-C/Hi–C contact maps with multiple public ChIP-Seq datasets (Fig. 3a). Two adjacent genes demonstrated three possible combinations of orientations (or directions for RNA Pol-mediated transcriptional elongation) of the gene pairs: divergent, convergent, and tandem (Fig. 3a, b). Both divergent and tandem transcription exhibited strong insulation with a distinct boundary separating two local chromatin domains (Fig. 3a); however, convergent transcription did not reveal a clean boundary (Fig. 3a). These patterns were correlated with the RNA Pol-mediated transcriptional elongation and H3K36me3 activity–closely associated with transcriptional elongation–rather than the activities of other histone modifications (Fig. 3a). Another example of gene transcription and local chromatin domains originated from the convergent transcription of two adjacent PCGs (*HISTONE DEACETYLASE 4* [*HDT4*] and *UDP-D-APIOSE/UDP-D-XYLOSE SYNTHASE 1* [*AXS1*]); it revealed distinct boundaries related to the binding of a tRNA gene by RNA Polymerase III (Pol III) (Fig. 3b). Moreover, we further compared the locations of tRNA genes and chromatin boundaries and found a significant overlap between them (the number of tRNA genes located in chromatin boundaries was 266, accounting for 43% of the total tRNAs (permutation test, *p*-value < 0.001). Indeed, Aggregate TAD analysis (ATA) revealed very similar behavior (e.g., strong boundaries in divergent promoters, moderate boundaries in convergent promoters, and weak boundaries in tandem promoters) in both yeast and *Arabidopsis* in the context of three types of gene pairs (Fig. 3c), consistent with conclusions based on a single locus (Fig. 3a, b).

Transcriptional activity is positively and negatively correlated with chromatin compaction in mouse cells[5] and yeast[6], respectively. Here, we assessed the relationship between transcriptional activity and chromatin in *Arabidopsis* by ATA across nine groups of PCGs, ranging from silenced genes (1) to highly expressed genes (9) (Supplementary Fig. 5). We found that stronger chromatin domains and boundaries were present in genes with high gene expression levels and elevated Pol II occupancy levels, based on analyses of unscaled and scaled genes (Fig. 3d, e and Supplementary Fig. 5). Completely silent genes were not located near very sharp boundaries, and chromatin interactions across

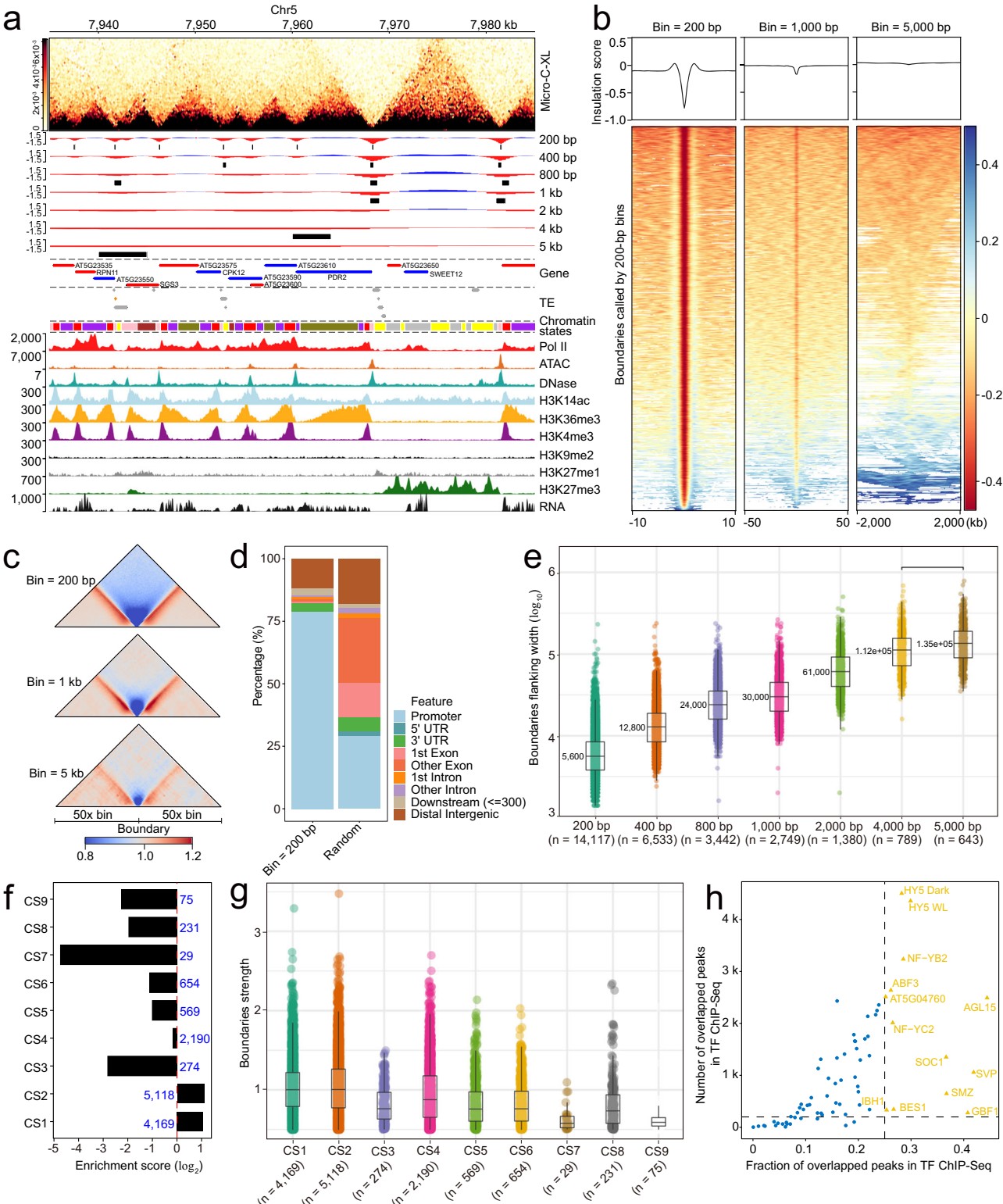

genes were weak (Fig. 3d, e). This observation is consistent with previous findings that the 5' and 3' ends of genes can form small-scale chromatin loops (i.e., gene loops) in *Arabidopsis*. These gene loops are more likely to originate from highly expressed genes[16]; however, a dot-like chromatin structure may not form at each end of all genes (Fig. 3e). We confirmed that these gene loops are more often present in gene "crumples" or globules, as defined in yeast (Fig. 3e).

Together, these results suggest that gene transcription (i.e., RNA Pol elongation) correlates with local chromatin organization.

## Pol II can affect gene-level chromatin domains

To determine whether changes in Pol II chromatin occupancy contribute to local gene-level chromatin organization, we designed a system that included two independent methods to knockdown Pol II; thus, we were able to further assess the role of Pol II in local gene folding (Supplementary Fig. 6a and Methods). We made use of a hypomorphic mutant[38] deficient in NRPB2 (which is the second-largest subunit of nuclear RNA polymerase B) and applied flavopiridol (FVP) exogenously to WT *Arabidopsis* seedlings, which specifically inhibits

**Fig. 2 | Chromatin boundaries separate the genome into distinct blocks. a** Locus showing the insulation score identified boundaries and local chromatin organization patterns. Micro-C-XL heatmap demonstrating chromatin interactions at 200-bp resolution. Insulation score distributions at multiple resolutions are shown. For each resolution, boundaries identified from the interaction matrix are indicated by black boxes. Tracks indicating PCG annotations, TE annotations, chromatin states, ATAC-Seq and DNase-Seq, Pol II occupancy, multiple epigenetic modifications (e.g., H3K14ac, H3K36me3, H3K4me3, H3K9me2, H3K27me1, and H3K27me3), and RNA expression are presented under the insulation score tracks. **b** Heatmap of insulation scores at three different resolutions (200-bp, 1000-bp, and 5000-bp) over boundaries identified in 200-bp resolution Micro-C-XL matrixes with larger flanking regions. **c** Heatmap showing aggregated chromatin interactions at different resolutions across boundaries from the same 200-bp resolution Micro-C-XL matrixes with adjacent 50-fold regions relative to each resolution. **d** Genomic distributions of boundaries in different regions of *Arabidopsis* genome relative to randomly shuffled regions. **e** Boxplot showing length distribution of flanking regions of boundaries identified by Micro-C-XL matrixes at different resolutions. Median values are shown near boxplots. The n value represents the number of flanking regions of boundaries. **f** Bar plot showing relative enrichment scores of boundaries located in different chromatin state blocks. Raw overlapped numbers are superimposed on bars. **g** Boxplot showing distributions of boundary strengths for each border in different chromatin states. The *n* value represents the number of boundaries distributed in each CS. **h** Scatterplot showing the number and proportion of overlaps between boundaries and transcription factors based on publicly available ChIP-seq and ChIP-on-chip data. The x-axis shows the fraction of peaks that overlap with chromatin boundaries for a given TF, and the y-axis shows the absolute number of overlaps. For each block diagram in (**e**) and (**g**), the centerline represents the median value; the boxes show the 25th and 75th percentiles; and the last of the whiskers represent the minimum and maximum values. Source data are provided as a Source Data file.

---

transcriptional elongation of Pol II[39]. Both treatments (named separately as *nrpb2–3* and FVP) resulted in severe developmental phenotypes (Supplementary Fig. 6b, c). To further investigate the manner by which a defect in Pol II activity affects chromatin structure at the molecular level, we conducted three types of experiments (RNA-Seq, Pol II CTD ChIP-Seq, and Micro-C-XL), each with two independent biological replicates, which yielded good quality data (Supplementary Fig. 7a–c). Large-scale analysis at the genome-wide level at 100-kb resolution demonstrates that the reduction in Pol II activity caused obvious changes in the global chromatin organization (Supplementary Fig. 7d–g), which is different from the minimal overall impact of the FVP inhibitor on mouse embryonic stem cells (ESCs)[5].

To further analyze the direct effect of a reduction in Pol II activity, we conducted an in-depth analysis using single genes at nucleosome resolution. It was revealed that the reduction in Pol II chromatin occupancy caused obvious changes in chromatin structure, and both treatments resulted in a similar molecular phenotype (Fig. 4a, b and Supplementary Fig. 7h, i). We screened gene loci with high levels of Pol II binding (~top 10%) in the WT and an obvious reduction in Pol II binding in *nrpb2–3* or FVP using genome-wide analysis (Supplementary Data 4; to avoid the impact of gene length, we only considered genes of medium length in the 2–3 kb range) following the recently published method[37]. We found dramatic changes in the chromatin occupancy levels of Pol II at these gene locations, which also affected the distribution of the insulation score calculated from individual Micro-C-XL (Fig. 4c, d). To accurately reflect the gene-level change in interaction intensity at an overall level, we still used the pileup method by scaled genes, which indicated that genes with weakened Pol II binding loosen the strength of chromatin structure (Fig. 4e). Moreover, from a statistical perspective, we further calculated the correlation between the fold change in Pol II chromatin occupancy level and chromatin intensity in *nrpb2–3* and FVP relative to WT. The results reveal a positive correlation (Fig. 4f) and indicate that the greater the change in Pol II chromatin occupancy, the more obvious the decrease in interaction intensity of chromatin structure at the single-gene level.

In summary, using two different systems at the single-gene level, we demonstrate that Pol II can directly regulate local chromatin structure.

### Chromatin domains associated with epigenetic modification

In addition to the close associations of local chromatin domains with gene transcription and RNA Pols, we found that multiple chromatin interaction domains were closely associated with epigenetic modification (Fig. 5a, b). Nucleosome-resolution Micro-C-XL mapping revealed that many chromatin interaction domains were associated with these repressive modifications (H3K27me3 and H3K9me2/H3K27me1), which were clearly visible despite the small size of this highly histone-modified region (Fig. 5a, b). Our results support previous findings that some large regions occupied by H3K27me3/H3K9me2 constitute local interactive domains or compartmental domains[16,20,21,40,41]. In addition to the close associations of domains with inactive modifications, we identified a previously unreported class of interactions by Micro-C-XL mapping (Fig. 5c). These are presumably specific interactions between intergenic regions and promoters and/or gene bodies and have no apparent relationship with known common histone modifications (Fig. 5c).

Furthermore, we identified some previously reported genes with chromatin loops or chromatin domains. For example, abundant chromatin interactions were present at the *FLC* site, from the promoter region to the region downstream of *FLC*; these regions were extensively occupied by H3K27me3 modifications (Fig. 5d), consistent with a previous report[26]. Chromosome conformation capture (3C) revealed that the promoter and TSS of *FLC* contacted its downstream regions, thus forming a chromatin loop; this loop formation was positively correlated with *FLC* expression activity[26]. Micro-C-XL observed/expected contact intensity mapping revealed that stronger chromatin interactions were generally present between the 5′ and 3′ ends of *FLC* (Fig. 5d). A similar example about H3K27me3-associated chromatin domain was also observed at *GLABRA2* (*GL2*) (Fig. 5e). Chromatin reorganization across *GL2* is required to regulate cell fate decisions in root hair cells upon receipt of positional information[24]. Accordingly, we identified H3K27me3-mediated chromatin compaction across the promoter and intragenic regions of *GL2* (Fig. 5e).

### Chromatin loops/stripes link SE to its cognate target gene(s)

To further characterize these interaction patterns (Fig. 5c), we integrated the chromatin accessibility data from the transposase-accessible chromatin assay using sequencing (ATAC-Seq) and DNase-Seq with the Micro-C-XL mapping data (Figs. 5c and 6a). We found that these chromatin interactions occurred between clustered open chromatin regions and adjacent promoters or gene body regions of individual PCGs (Figs. 5c and 6a, b). It has been suggested that open chromatin regions in *Arabidopsis* can serve as enhancers[28] and/or SEs[29]; however, C-series technologies (e.g., Hi–C, Capture-C, and 4C using restriction enzymes) are limited in terms of analyzing interactions in small regions that include enhancers and nearby genes. Micro-C-XL is appropriate for the combined analysis of chromatin interactions and SEs or enhancers.

In combination with the 749 recently published SEs in *Arabidopsis*[29], the longer length of the SEs in our study facilitated the integration of their 1D locations and 3D interactions from the Micro-C-XL map. To fully analyze the relationships between SEs and these specific chromatin interactions, we sequentially examined the chromatin interaction maps of Micro-C-XL and Hi–C within regions of approximately 50 kb, both upstream and downstream of each SE (Fig. 6a–c and Supplementary Fig. 8). Analysis of associated chromatin interaction patterns revealed ≥ 4 types of SEs (Fig. 6a–d and Supplementary Fig. 8). The first SE type (i.e., loop) can form clear dot-like

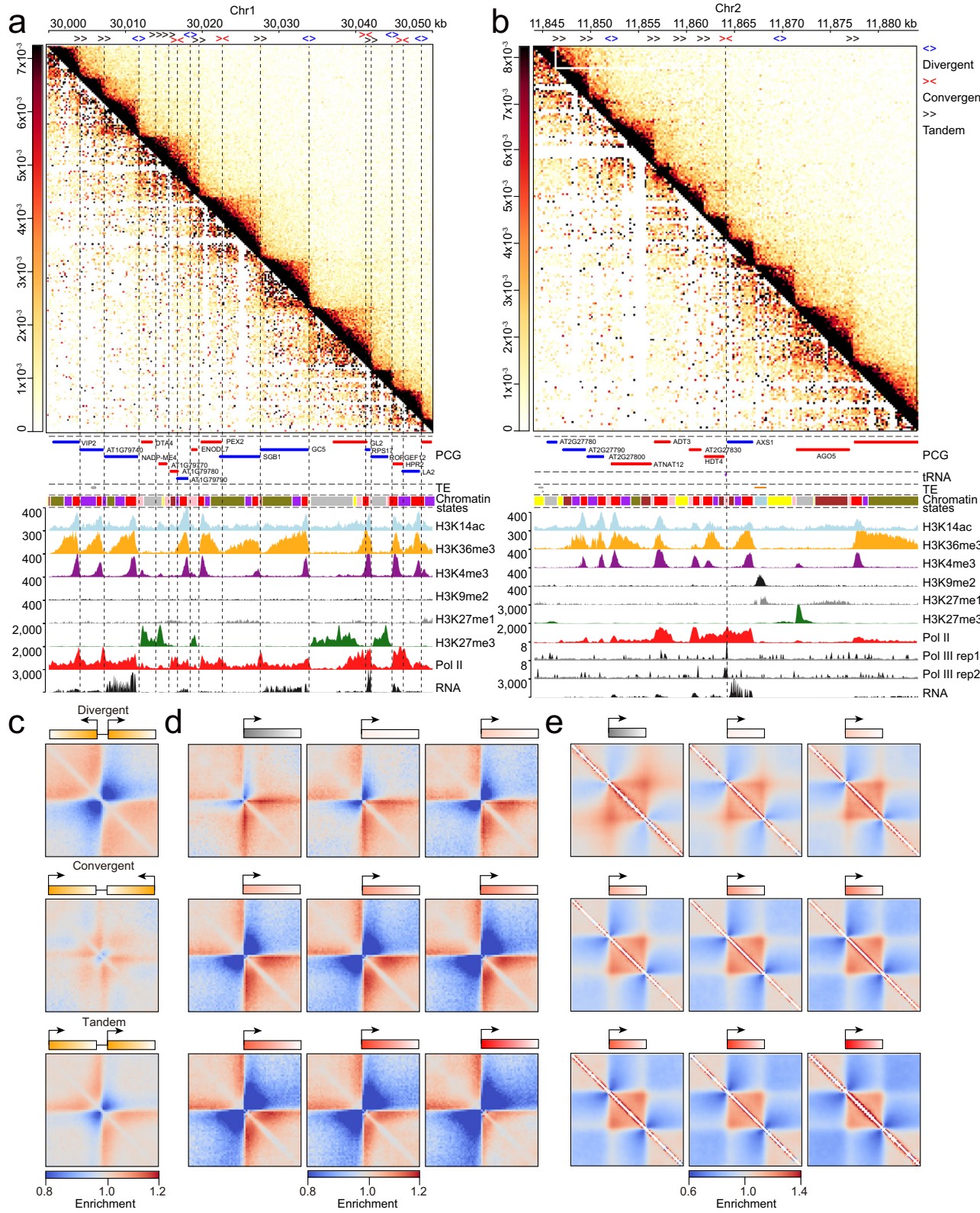

structures between an SE and the promoter or gene body of its target gene. For example, a distinct chromatin loop is formed between a large upstream SE and *RELATED TO ABI3/VP1* (*RAV1*). Similar situations are present in other genes, including *AT1G29640*, *RELATED TO AP2 4* (*RAP2-4*), and *ETHYLENE RESPONSIVE ELEMENT BINDING FACTOR 4* (*ERF4, also known as RAP2.5*) (Supplementary Fig. 8a–c). The second SE type is similar to the loop type but interacts in a different manner through the formation of chromatin stripes, which are asymmetrical

and band-like structures. An example involves the interaction between an SE and *AUXIN RESPONSE FACTOR 19* (*ARF19*) (Fig. 6b). Similar stripes are present in other genes, including *AT1G13350* and *KNOTTED1-LIKE HOMEOBOX GENE 3* (*KNAT3*) (Supplementary Fig. 8d, e). The third SE type is located in a specific domain; for example, the *AUXIN RESISTANT 1* (*AUX1*) gene and its nearby SE are located in an H3K27me3-tagged domain (Fig. 6c), consistent with a previous report[29]. The fourth SE type includes SEs that remain difficult to assign based on specific

**Fig. 3 | Transcription is closely associated with the local presence of a triangle chromatin domain. a** Asymmetrical heatmap showing chromatin interactions in Hi–C (left lower corner) and Micro-C-XL at 200-bp resolution (upper right corner). Tracks showing PCG annotations, TE annotations, chromatin states, multiple epigenetic modifications (e.g., H3K14ac, H3K36me3, H3K4me3, H3K9me2, H3K27me1, and H3K27me3), Pol II occupancy, and RNA expression are presented under chromatin interactions heatmap. Vertical dashed lines are displayed on maps and tracks to facilitate chromatin organization, gene transcription, and gene arrangement analyses. Three gene arrangements, including divergent (blue arrow), convergent (red arrow), and tandem (black arrow) gene pairs, are presented at the top of the map. **b** Similar to **a**, with the addition of tRNA gene annotations and Pol III occupancy data. One dashed line was added, showing the location of a tRNA gene. **c** Aggregated analysis of chromatin interactions over three types of gene arrangements. Ideograms showing arrangements are presented at the top of each heatmap. **d** Aggregated analysis of chromatin interactions using transcription start sites (TSSs) from nine types of genes with increasing expression levels (greater expression is indicated by greater red intensity, except in first class [i.e., silenced genes]). **e** Aggregated analysis of chromatin interactions using gene bodies from nine types of genes with increased expression levels, using ideograms presented in (**d**). Enrichment scores are relative to a random background.

interactions. Various SEs are distributed among fine-scale chromatin interactions, implying that different SE-associated chromatin interactions contribute to distinct underlying mechanisms (Fig. 6d). However, the tissue/organ/cell-type specificity of chromatin organization patterns must be considered. Thus far, we have generated a Micro-C-XL map for the seedling stages of *Arabidopsis*, although SEs are derived from multiple tissues and ecotypes. In the future, we expect that additional Micro-C-XL data will be acquired under different conditions; this information will facilitate analyses of the underlying mechanisms of SEs.

## Identification and analysis of possible key regulators of chromatin loops

To identify factors potentially involved in specific interactions with SEs, we tested ChIP-Seq of multiple factors associated with epigenetic and transcriptional regulation. We found that mediator 12 (MED12), microrchidia 7 (MORC), and the SYD-associated SWItch/Sucrose Non-Fermentable (SWI/SNF) (i.e., SAS) complexes, but not Pol II and H3K27me3, were enriched across SEs (Supplementary Figs. 9 and 10). The core members of SAS complexes (e.g., SYD, SWI3D, and SYS123) can bind to intergenic regions and directly regulate the degree of chromatin opening in distal intergenic regions[42], suggesting that SAS complexes directly participate in the formation of chromatin loops; they also participate in the regulation of remote genes (Supplementary Fig. 10). Thus, we plan to establish a model through which open regions distant from genes can be identified by the integration of ATAC-Seq/DNase-Seq, thereby enabling the prediction of potential enhancers and SEs. The integration of ultra-resolution Micro-C-XL technology will also facilitate the identification of potential target genes for enhancers/SEs, along with possible mechanisms of enhancer–promoter action. Moreover, MORC7, mediator complexes, and SAS complexes are potential regulators of these SEs and SE-associated chromatin organization patterns (Fig. 6e).

To further elucidate the relationship between 1D data for specific regulatory factors and 3D data such as chromatin loops, we simultaneously analyzed the ChIP-Seq data for the typical factors described above and the Micro-C-XL data. The *Arabidopsis* chromatin contact map indicates that most of the chromatin loops/stripes are very short, mostly within the 50-kb range, and display varying patterns with strong heterogeneity. To explore this problem, based on nucleosome-resolution contacts and public ChIP-Seq peaks, we designed a scheme to quantitate the average potential of a specific protein to participate in the chromatin loops at the whole-genome level using aggregate peak analysis[43] (APA) and paired-end spatial chromatin analysis[44] (PE-SCAn) (Fig. 7a). We used an existing peak called by certain protein ChIP-Seq to search all possible interaction regions within the 50-kb range (to prevent dilution signals from the 2D regions far away from the diagonal and control the search space) and then took an average to determine the potential of a factor being a looping factor (Fig. 7a). Using this method, we tested the looping potential of the factors, including the enhancers of the leaf (as a control), MORC6/7, NRPE1 (previous studies have reported widespread co-localization between MORC6/7 and Pol V (NRPE1)), MED12, and SYD (Fig. 7b). We also specifically analyzed MED12 and SYD and uncovered their potential to participate in the formation of chromatin loops; however, the overall mode is very different from that mediated by MORC/Pol V (Fig. 7b).

Based on previous studies on MORC family proteins[21,45,46], we hypothesized that MORC may be a molecular motor directly involved in the formation and/or maintenance of chromatin organization. By combining these asymmetric heatmaps with various ChIP-Seq data, we found that there are both chromatin loops strongly associated with only MORC proteins and with both MORCs and Pol V (Fig. 7d, e).

These results show that MORC6/MORC7/NRPE1 and other proteins are potentially involved in chromatin loops (Fig. 7b), which is consistent with a previous report that MORC6 may be a chromatin motor protein in plants. Nevertheless, the main focus has been its role in large-scale chromatin structure[21,45], which prompted us to focus on chromatin loops from the perspective of MORCs and NRPE1. The MORC family contains six active PCGs in *Arabidopsis*, which limits the direct detection of chromatin structural changes in mutants of MORCs[47]. It is noteworthy that the Pol V arm of the RNA-directed DNA methylation (RdDM) pathway co-localizes with the MORC4/7 binding site and has been shown to recruit MORC7[48]. Therefore, we changed our perspective and carried out Micro-C-XL experiments and analyses in RdDM mutants, including *nrpe1–11* and *nrpd1–3* (Fig. 7b, c), which are the largest unique subunits of plant-specific RNA polymerase V and IV, respectively, yielding good quality data (Supplementary Fig. 11). The Pol V and Pol IV mutants slightly altered the large-scale chromatin organization (Supplementary Fig. 11) but the effect was weaker than that of the Pol II series (*nrpb2–3* and FVP). Using APA analysis, we found that Pol IV and Pol V slightly affected the interaction strength between leaf enhancers; unfortunately, these Pols had no effect on the chromatin loops associated with MORC6/7 or Pol V itself. These data suggest that the formation and maintenance of short chromatin loops are also uncoupled, which is consistent with previous studies demonstrating the relationship between RdDM and MORC4/7[48]. Genetically, RdDM can recruit MORC7, but the maintenance of MORC7 is not dependent on RdDM.

From another perspective, although Pol IV and Pol V do not affect the short-range chromatin loops, they greatly impact the local chromatin structure over the Pol V-bound regions (Fig. 7c, f). This is consistent with the fact that the Pol V complex tends to stick to chromatin rather than only producing RNA molecules, which is quite different from Pol II[49].

In summary, we established an effective strategy to analyze chromatin loops mediated by specific proteins and identified proteins such as MORC as possible factors. By analyzing RdDM mutants, we found that Pol V, a plant-specific RNA Pol, simultaneously participates in local domain and short-range chromatin loops without affecting them, which further indicates that RNA Pols play an important role in the composition of chromatin structure in plants.

## Application of Micro-C-XL to other plants

To confirm that Micro-C-XL can be widely used in plants and characterize its advantages in terms of dissecting fine-scale chromatin organization patterns, we constructed Micro-C-XL libraries in rice and soybean (two biological repeats were included for each species) (Fig. 8a, b). More than 400 million valid interactions were identified in

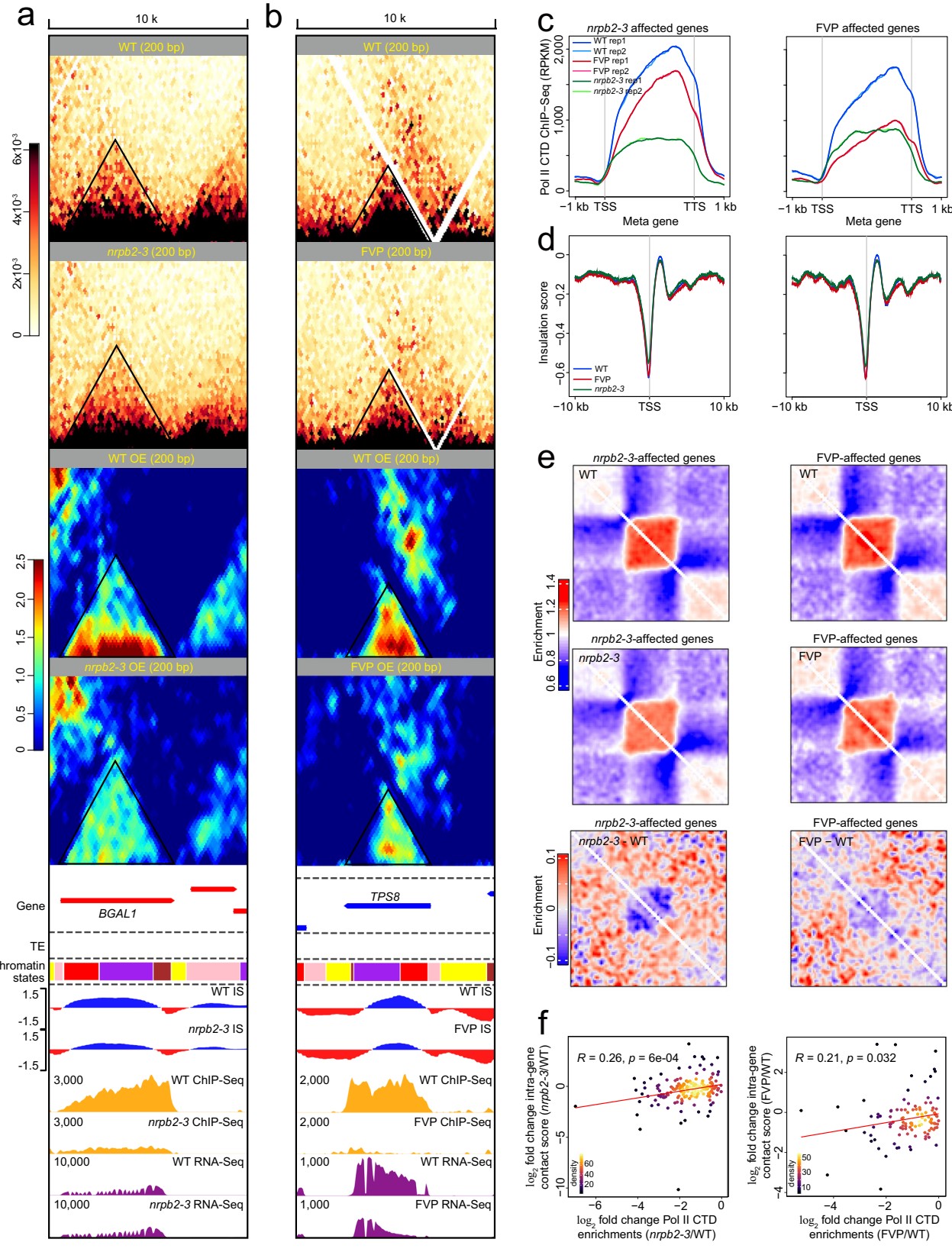

each Micro-C-XL library (Supplementary Data 1). The scaling plot indicated good reproducibility (Supplementary Fig. 12a, b). *Cis*/total ratios of the four libraries exceeded 80%, indicating that Micro-C-XL can be successfully used for analyses of both species (Supplementary Data 1). The effective ratios relative to raw sequencing read pairs were also high (Supplementary Data 1).

In both rice and soybean, Micro-C-XL detected chromatin organization patterns that had been identified by Hi−C (Fig. 8a, b). In comparison with Hi−C, Micro-C-XL revealed more detailed chromatin organization in rice at 400-bp resolution, even when only two libraries were used (Fig. 8a). Similarly, Micro-C-XL exhibited clear advantages over Hi-C at 800-bp resolution in soybean (Fig. 8b). To further

**Fig. 4 | Pol II affects local chromatin organization at the single-gene level.**
**a** Horizontal contact heatmap showing normalized chromatin interactions in WT and *nrpb2–3* Micro-C-XL at 200-bp resolution. A black triangular box indicates a typical chromatin domain containing the entire single gene. Tracks showing PCG annotations, TE annotations, chromatin states, and insulation scores from Micro-C-XL, Pol II CTD ChIP-Seq, and RNA-Seq in WT and *nrpb2–3* are displayed underneath. IS, insulation score. **b** Similar to **a**, using FVP datasets rather than *nrpb2–3* and displaying another locus. **c** Metagene analyses of Pol II CTD chromatin occupancy over chosen genes in WT, *nrpb2–3*, and FVP (left, chosen *nrpb2–3*-affected genes from *nrpb2–3* vs. WT; right, chosen FVP-affected genes from FVP vs. WT). Genes with the top 10% binding intensity of Pol II CTD in WT and those with lower Pol II CTD binding in *nrpb2–3*/FVP vs. WT were chosen. Chosen genes had a length

ranging from 2 kb to 3 kb to avoid impacts of gene length. DMSO-affected genes were excluded from the analysis of FVP vs. WT. **d** Metagene analyses of the insulation scores calculated from WT, *nrpb2–3*, and FVP Micro-C-XL across chosen genes (left, *nrpb2–3*-affected genes; right, FVP-affected genes). **e** ATA of chromatin interactions in WT and *nrpb2–3*/FVP using the chosen gene sets. Scores from *nrpb2–3* vs. WT and FVP vs. WT were plotted as the ATA difference maps. Enrichment scores are relative to a random background. **f** Linear regression analysis. The x-axis represents the log$_2$ ratio of Pol II binding intensity at each gene from *nrpb2–3* relative to WT, and the y-axis represents the log$_2$ ratio of chromatin interaction intensity at each gene from *nrpb2–3* relative to WT. The left scatterplot represents *nrpb2–3* vs. WT; the right scatterplot represents FVP vs. WT. Source data are provided as a Source Data file.

illustrate these advantages, we generated ATA plots for previously identified domains in rice and for soybean TADs identified in our laboratory. Micro-C-XL ATA plots of rice domains and soybean TADs were both superior to Hi–C plots (Fig. 8c, d), suggesting that Micro-C-XL is more appropriate for analyses of fine-scale chromatin organization, even with the same number of valid interactions. In summary, we successfully performed Micro-C-XL analyses in rice and soybean and confirmed the superiority of Micro-C-XL relative to Hi–C.

## Discussion

In this study, we successfully used Micro-C-XL for unbiased mapping of ultra-fine resolution higher-order chromatin organization in *Arabidopsis*, then demonstrated the superior performance of Micro-C-XL technology (compared with traditional Hi–C technology) for analysis at nucleosome resolution. Micro-C-XL provided a clear understanding of chromatin domains and their boundaries in *Arabidopsis*; our findings indicate that gene transcription and epigenetic domains shape the local arrangement of the 3D genome. Furthermore, we identified a class of chromatin loops/stripes that form between SEs and their target genes, indicating that the combined use of Micro-C-XL with ATAC-Seq/DNase-Seq can provide robust analyses of fine-scale regulatory relationships between CREs and nearby genes. Finally, we successfully constructed high-resolution Micro-C-XL maps in rice and soybean.

The greatest strengths of Micro-C-XL are its ultra-fine resolution and comparatively lower signal-to-noise ratio relative to conventional technologies (e.g., Hi–C). For plant species with small genomes, such as *Arabidopsis*, two Micro-C-XL libraries (a library means a pool of DNA fragments with ligated adapters that can be directly sequenced by a specific platform derived from a Micro-C-XL experiment) are sufficient for nucleosome resolution. For crops with genomes of approximately 400 Mb (e.g., rice), four Micro-C-XL libraries should achieve near-nucleosome resolution; and 6 libraries should yield good results for crops with genomes of approximately 1 Gb (e.g., soybean). Thus, direct use of the Micro-C-XL library is sufficient for species with genomes smaller than 3 Gb (e.g., mammals). One of the main drawbacks of using Micro-C-XL is the high expense involved in constructing multiple libraries and sequencing for species with large genomes. Additionally, since next-generation sequencing is still utilized, the difficulty in analyzing transposons and highly repetitive sequences remains. Furthermore, the procedure for constructing a library is complicated and necessitates the careful calibration of MNase concentration. Concerning crops with an excessively large genome (e.g., ~16 Gb in wheat), improvements may be achieved by investigating promoter- and enhancer-centered interaction maps using Micro-C-XL technology supplemented with probe capture. Interaction maps with near single-base resolution can be generated by focusing on local interactions within small regions of the genome using Micro-Capture-C[8] and tiled-Micro-Capture-C[9]. Overall, the development and application of these technologies in plants are essential for analyzing interactions between CREs and promoters/genes, in addition to exploring their underlying mechanisms. Further evaluation of the biological functions

of higher-order chromatin structures and investigation of mechanisms regulating gene expression are needed. It is important to note that the arrangement of chromatin structure differs depending on the organ, tissue, or cell type[24,50]; thus, there is considerable interest in analyses of specific cell types using Micro-C-XL or even single-cell Micro-C, which should be considered in future studies on cell heterogeneity.

Our work and another recently published article[36] have found many chromatin boundaries in *Arabidopsis*. Different factors, such as H3K27me3 and H3K9me2, may mediate the compartmental domains around these boundaries[36]. The present study focuses on the Pol II series, which widely exists in the *Arabidopsis* genome. We found such local compartmental domains closely related to gene transcription and gene arrangements and further demonstrated that Pol II can regulate local chromatin domains by altering their binding. These results are highly consistent with the conclusion that a reduction in Pol II activity affects chromatin structure at the gene level in mammals[5,37]. We studied the relationship between Pol II and chromatin structure in plants instead of cell lines, offering an additional perspective and providing valuable insights for studying other protein factors and their biological functions in chromatin structural organization.

SE-mediated chromatin loops and strips were the most intriguing results of the present study; to our knowledge, they have not previously been identified in *Arabidopsis*. These findings, which are conceptually similar to the enhancer–promoter loop model in mammals[51], suggest that Micro-C-XL (coupled with 1D omics) can be used to identify the regulatory patterns and relationships between CREs (including enhancers, silencers, and other regulatory elements) and genes. Identification and characterization of specific chromatin loops and their molecular mechanisms and functions in plants are important current goals in 3D genomics.

Furthermore, there is a need to identify the proteins responsible for the punctate loops and stripes formed between these CREs, such as SEs, and their cognate genes. We found that some regulatory factors (e.g., MORC7, MED12, and SAS complexes) are potentially involved in establishing these chromatin structures. The MORC family has been reported to affect chromatin structure within the nucleus in plants by Hi–C[21,45] and biochemical evidence from studies in *Caenorhabditis elegans* suggests that MORC family members can compact chromatin[46]. Although MORC-mediated chromatin structure is unfortunately unaltered in Pol V and Pol IV mutants, our work presents an effective approach to identifying proteins that may control chromatin structure. It is crucial to discover the master proteins (similar to CTCF and YY1 in mammals) responsible for organizing chromatin structure (especially chromatin loops connecting CREs and targeted genes) in plants. Only then can we further understand the mechanism and function of high-order chromatin structure and the regulation of gene expression at the 3D level. We believe that integrating Micro-C-XL and related ChIP-Seq technologies is conducive to establishing a direct connection between the two at nucleosome resolution. Our Micro-C-XL assay has nucleosome resolution, allowing for the study of specific protein-mediated interactions, as evidenced by short peaks in ChIP-

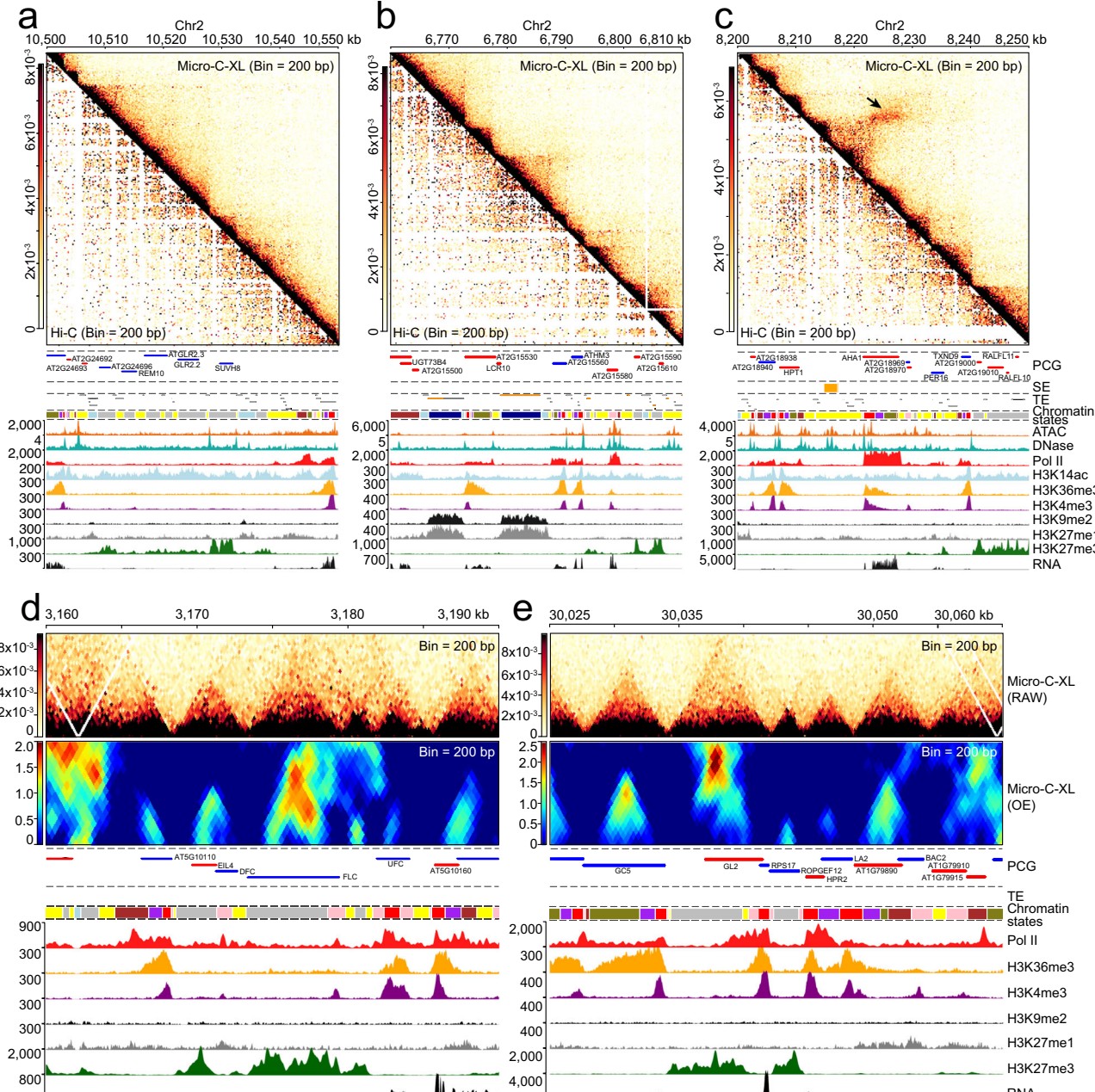

**Fig. 5 | Epigenomic features compose local chromatin domains. a** Typical locus showing a chromatin domain composed of H3K27me3. Asymmetrical heatmap showing chromatin interactions in Hi–C (left lower corner) and Micro-C-XL at 200-bp resolution (upper right corner). Tracks showing PCG annotations, TE annotations, chromatin states, ATAC-Seq and DNase-Seq, Pol II occupancy, multiple epigenetic modifications (e.g., H3K36me3, H3K4me3, H3K9me2, H3K27me1, and H3K27me3), and RNA expression are presented under the chromatin interaction heatmap. **b** Typical locus showing chromatin domain composed of H3K9me2/H3K27me1. Tracks of other annotations are identical to (**a**). **c** Typical locus showing chromatin interactions. Arrows label key interactions. **d** Typical locus showing chromatin interaction patterns over *FLOWERING LOCUS C* (*FLC*). Normalized contact intensity is shown at the top of the panel. The observed/expected (OE) matrix is shown on the next track. Other tracks are identical to (**a**). **e** Typical locus showing chromatin interaction patterns over *Glabra 2* (*GL2*). Other tracks are identical to (**a**).

Seq of protein factors. Based on our current work and previous studies, it appears that the MORC family may serve as important motor proteins in the regulation of chromatin structure; nevertheless, further genetic and biochemical evidence is still required. MED12 is widely regarded as a key regulator involved in enhancer–promoter loops, mediating gene activation effects[52–54]. Numerous reports have also demonstrated that mammalian chromatin remodeling complexes are key components in enhancer–promoter loops that activate gene expression[54,55]. Hi–C studies in mutant *Arabidopsis* have shown that chromatin remodeling complexes can regulate chromatin compartments by altering the distribution and density of nucleosomes while

promoting the deposition of H3K27me3[56]. Therefore, we speculate that these proteins directly regulate specific chromatin interactions between SEs and genes. There is a need for additional genetic evidence through Micro-C-XL analyses of plants with mutant forms of these proteins.

Finally, it is important to determine whether these chromatin loops and stripes are directly involved in regulating gene expression. Therefore, we need to go from *cis* and use gene editing to knock out these SEs or mutants with T-DNA insertions to disrupt SE structures to examine their expression change patterns for nearby potential target genes. Specifically, more nuanced manipulation of such SEs may be

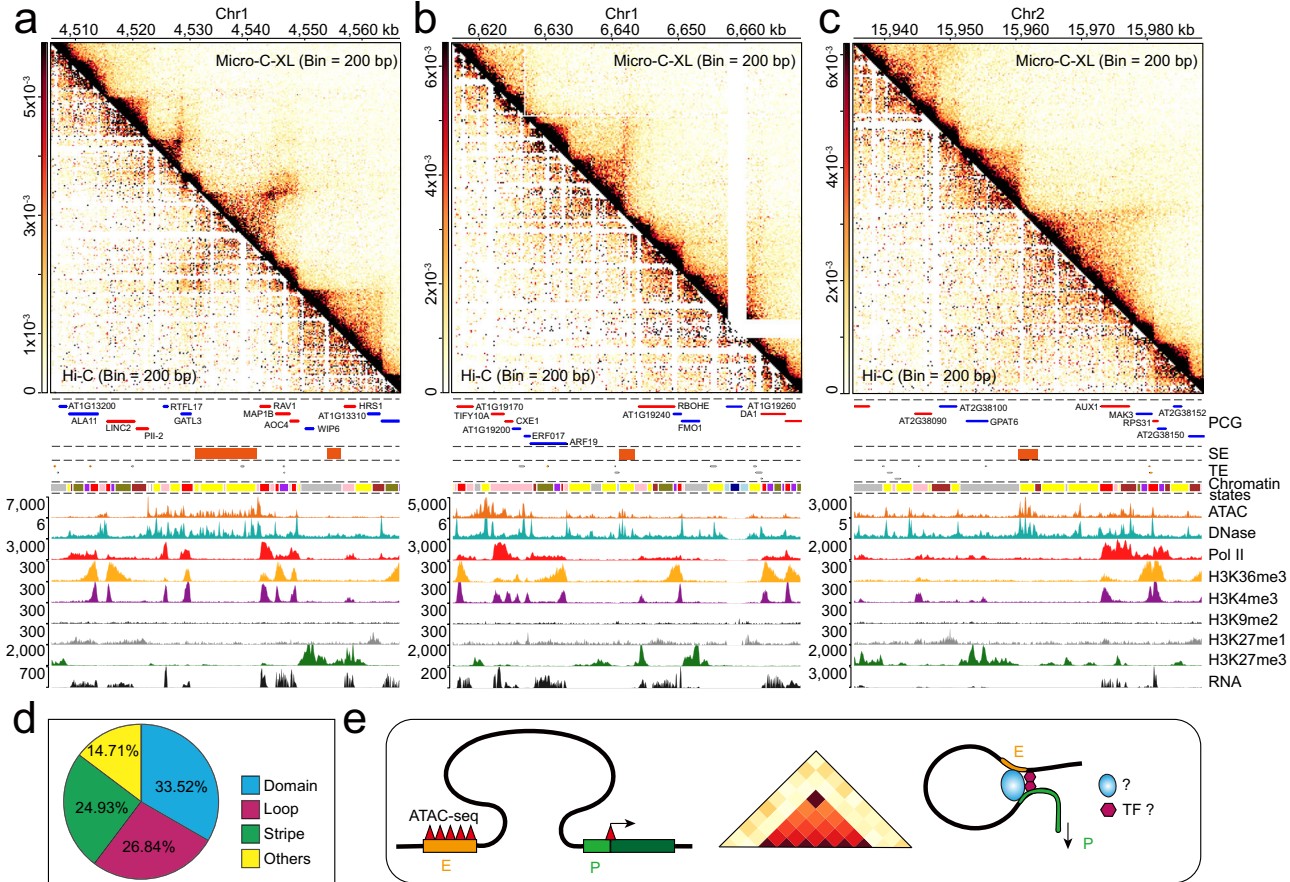

**Fig. 6 | Chromatin loops and/or stripes link super-enhancers to their potential target gene(s). a** Asymmetrical heatmap showing chromatin interactions in Hi−C (left lower corner) and Micro-C-XL at 200-bp resolution (upper right corner). PCG, super-enhancer, TE, and chromatin states are presented under the heatmap. ATAC-Seq and DNase-Seq tracks are presented, along with tracks shown in previous figures. Heatmap showing a typical locus containing a chromatin loop between an SE and *RAV1*. **b**, **c** Other types of chromatin organization patterns: stripe and domain. **d** Pie chart showing distributions of SE-associated chromatin organization patterns: loop, domain, stripe, and others (patterns without distinct structure). **e** Model showing that combined analyses of SEs (clustered distal open chromatin regions), performed by ATAC-Seq/DNase-Seq, and short-range chromatin loops/stripes from fine-scale chromatin organization patterns, performed by Micro-C-XL, can facilitate the identification and characterization of SEs and promoter interacting loops. Chromatin regulatory proteins and specific transcription factors may function synergistically to form chromatin loops and regulate target gene expression.

possible in the future using gene insertion and deletion tools that involve prime editing[57], along with programmable addition via site-specific targeting elements (PASTE)[58].

## Methods

### Plant materials and growth conditions

All *Arabidopsis*, soybean, and rice plants used in this study were in Columbia-0 (Col-0), Willimas 82, and *Oryza sativa* L. (Nipponbare) backgrounds. *Arabidopsis* seedlings were grown on half-strength Murashige and Skoog (1/2 MS) solid medium containing 1% sucrose (m/v) and 0.6% (for horizontally grown seedlings) plant agar (m/v) at 22 °C in an incubator under full white light conditions (HiPoint FH-740, 60% light intensity). Seven-day-old seedlings were harvested for experiments. Soybean seedlings were grown on Gamborg B-5 (B5) solid medium containing 2% sucrose (m/v) and 0.25% (for horizontally grown seedlings) phytagel (m/v) in an incubator at 26 °C under full white light conditions (24 h of light (Percival ARC-36L2-E; 40% light intensity). Roots of 5-day-old seedlings were harvested for experiments. Rice seeds were germinated in a dark incubator at 30 °C in a culture dish containing sterile water. Three-day-old seedlings were transferred to a growth chamber at 28 °C. Under long sunlight (16 h of light [light-emitting diode, 50 W/Flat lamp-2 spectrum] and 8 h of darkness), seedlings were grown in a hydroponic incubator containing sterile water. Roots of 10-day-old seedlings were harvested for experiments.

*nrpb2−3* was a kind gift from Dr. Binglian Zheng (Fudan University). *nrpd1−3* (Salk_128428) and *nrpe1−11* (Salk_029919)[59,60] were provided by Dr. Weiqiang Qian (Peking University). For Pol IV and Pol V experiments with *Arabidopsis* seedlings, surface-sterilized seeds (using 75% ethanol) were sown on 1/2 MS plates containing 1% sucrose (m/V) and 0.6% (for horizontally grown seedlings) or 1.1% (for vertically grown seedlings) plant agar (m/V) (pH = 5.8), stratified at 4 °C for 2 d, and then grown in a growth chamber at 22 °C under long-day conditions (16 h light-8 h dark cycle). The aerial parts of 2-week-old seedlings were harvested for further experiments (see Supplementary Fig. 6a).

For morphological observation of seedlings or pharmacological experiments, *Arabidopsis* seeds (wild-type Col-0 and *nrpb2−3*) were sown on 1/2 MS plates containing 1% sucrose (m/V) and 1.1% plant agar (m/V). After stratification for 2 d at 4 °C, the plates were transferred to a growth chamber for 7 d. Then, parts of the WT (7 d) and *nrpb2−3* (7 d) seedlings were transferred to 1/2 MS plates containing 1% sucrose (m/V) and 1.1% plant agar (m/V). The remaining WT seedlings (7 d) were transferred to 1/2 MS plates containing 10 µM DMSO (Macklin, D806645-500 mL) or Flavopiridol (FVP, Selleck, S1230) for an additional 7 d. Finally, the aerial parts of 14-day-old seedlings were harvested for further experiments.

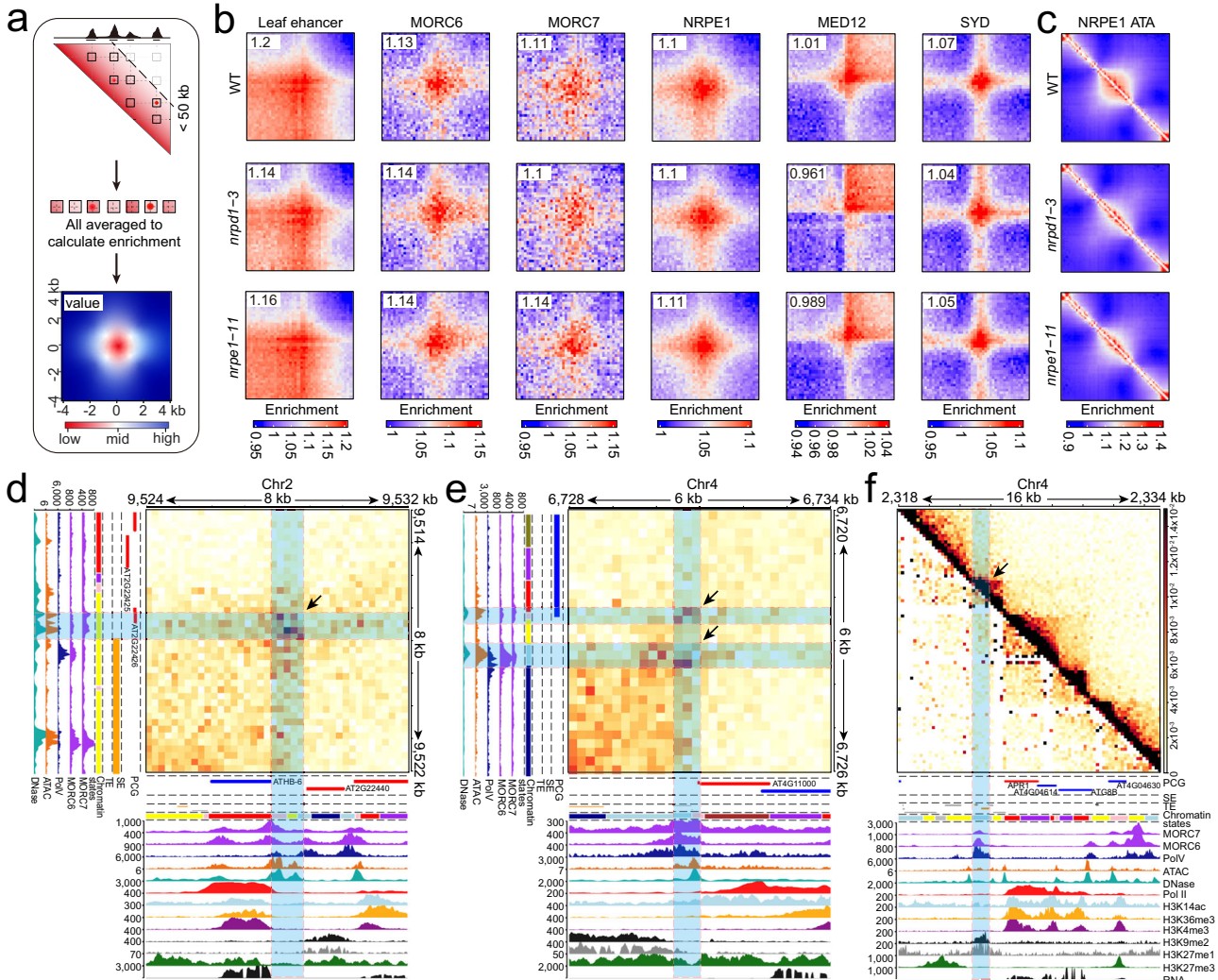

**Fig. 7 | Identification of possible protein factors involved in chromatin loops.**
**a** Schematic diagram showing the loop factor identification pipeline. The aim was to establish a link between certain 1D ChIP-Seq data and 3D Micro-C-XL data, quantitating the potential of a certain protein factor to participate in the chromatin loop. The pipeline only searched for 2D areas within the 50-kb range in which any two peaks interacted with each other and then extracted them for averaging. The heatmap shows a typical pileup of looping factors. Both horizontally and vertically, the length of the heatmap is only 8 kb. The interaction value in the central region is the highest and gradually decays toward the surroundings, which is a typical pattern for a loop. **b** Looping potential heatmap for leaf enhancers and other protein factors in WT, *nrpd1–3*, and *nrpe1–11*. **c** ATA heatmaps for NRPE1 (Pol V)-occupied regions in WT, *nrpd1–3*, and *nrpe1–11*. **d** Asymmetric heatmap shown with the Micro-C-XL contact map from two regions at 200-bp resolution (8 kb × 8 kb). The order from the main heatmap body downwards is PCG annotations, TE annotations,

SE annotations, ChIP-Seq for protein factors including MORC6, MORC7, and NRPE1 (the same with Pol V), ATAC-Seq and DNase-Seq, Pol II occupancy, multiple histone modification ChIP-Seq including H3K14ac, H3K36me3, H3K4me3, H3K9me2, H3K27me1, and H3K27me3, and finally RNA expression. On the left side of the heatmap body is simplified information, including PCG annotations, TE annotations, SE annotations, ChIP-Seq for protein factors including MORC6, MORC7, and NRPE1 (the same with Pol V), ATAC-Seq and DNase-Seq. **e** Typical examples for another locus (6 kb × 6 kb). All heatmaps and tracks are the same as in (**d**). **f** Symmetric contact heatmap showing a typical locus of *intergenic noncoding regions 5* (*IGN5*) targeted by Pol V. All annotations are the same as in (**d**) and (**e**). Shaded strips highlight the areas of specifically targeted chromatin interactions and the key protein binding peaks of ChIP-Seq from their two interacting anchors. Arrows label key interactions.

## Micro-C-XL experiment for plants

Micro-C-XL libraries were constructed in accordance with established methods[5,7] with three minor modifications. Firstly, we adjusted the composition of the MB buffer to make it suitable for plants (replacing Tris HCl pH = 7.5 with Tris HCl pH = 8.0 and 0.2% NP-40 with 0.1% Triton™ X-100). Secondly, following the reversal of crosslinking, we used the Qiagen DNeasy® Plant Mini Kit to extract DNA instead of using phenol:chloroform:isoamyl alcohol extraction. Thirdly, we used 0.7X + 0.3X Ampure XP beads to select and purify DNA of the target fragment length (250–400 bp) instead of agarose gel extraction. Briefly, ~2 g plant tissue was ground into a powder in liquid nitrogen and then incubated with Nuclei Isolation Buffer lysis solution

(20 mM HEPES pH = 8.0, 250 mM sucrose, 1 mM $MgCl_2$, 5 mM KCl, 40% glycerol, 0.25% Triton™ X-100, 1× β-mercaptoethanol, 1× phenylmethylsulfonyl fluoride, and 1× Roche complete ethylenediaminetetraacetic acid [EDTA]-free) at 4 °C for 15 min. After filtration through a 40-μm cell strainer, nuclei were collected by centrifugation and resuspended in cold 1× phosphate-buffered saline. Subsequently, nuclei were crosslinked for 15 min with 3% FA at 4 °C and then quenched with glycine (0.375 M final concentration) for 5 min. FA-fixed nuclei were subjected to additional cross-linking with freshly prepared 3 mM EGS for 60 min at 4 °C. FA + EGS dual crosslinked nuclei were quenched with 0.4 M glycine for 15 min at 4 °C, then washed twice with phosphate-buffered saline supplemented with 0.05% bovine serum

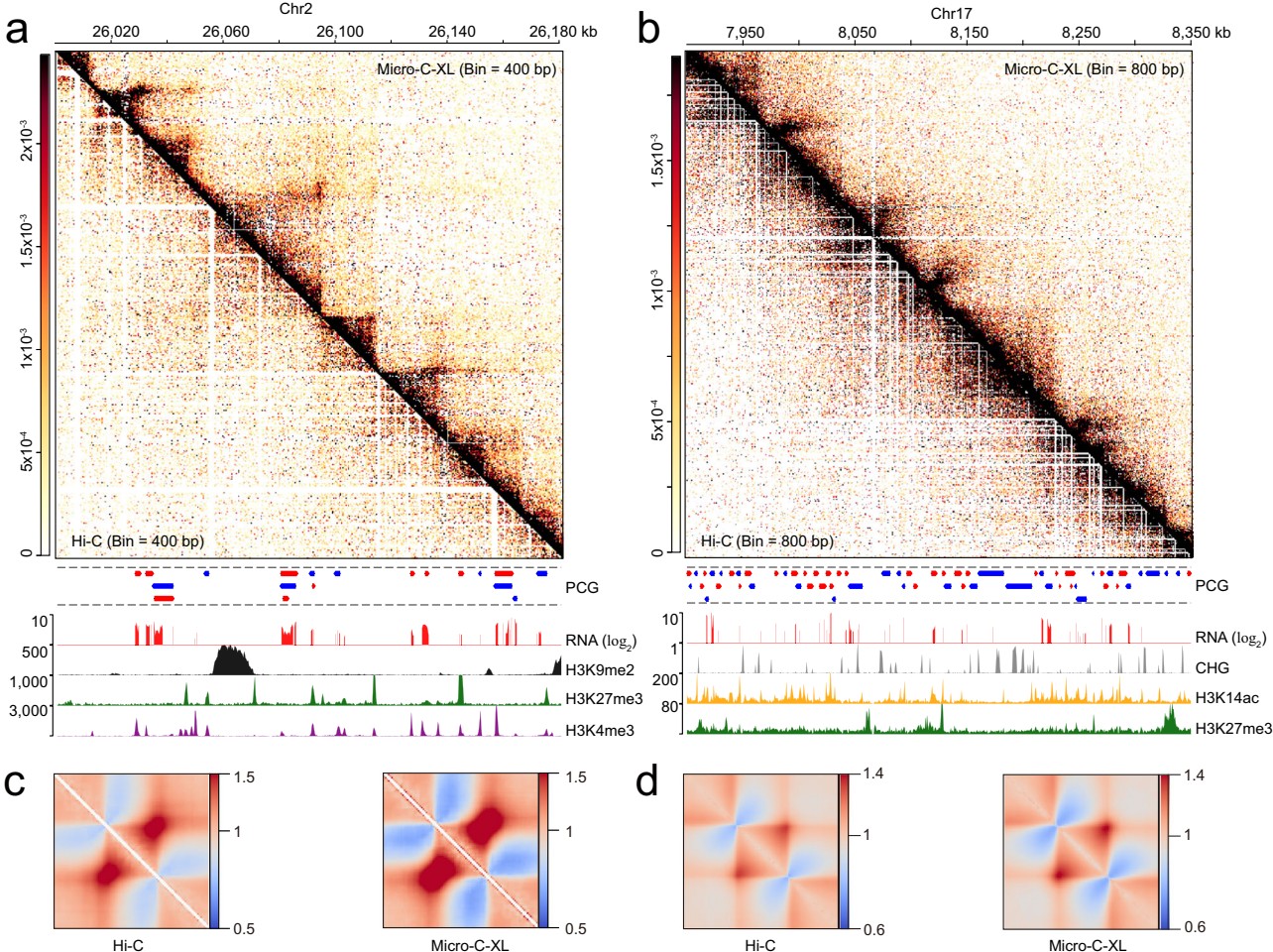

**Fig. 8 | Micro-C-XL outperforms Hi−C in rice and soybean. a** Asymmetrical heatmap showing chromatin interactions in Hi-C (left lower corner, data from Dong et al., 2018) and Micro-C-XL at 400-bp resolution (upper right corner) with RNA, H3K9me2, H3K4me3, H3K27me3, and PCG tracks in rice. **b** Asymmetrical heatmap showing chromatin interactions in Hi-C (left lower corner, from[82]) and Micro-C-XL at 800-bp resolution (upper right corner) with RNA, DNA CHG methylation, H3K14ac, H3K27me3, and PCG tracks in soybean. **c** ATA plots showing chromatin contact enrichment across TAD were generated based on previously published research (Liu et al., 2017)[90] using Hi-C (Dong et al., 2018)[91], as well as Micro-C-XL generated in this study. **d** ATA plots showing chromatin contact enrichment between TADs called using Hi−C[82] and Micro-C-XL (this study) with 5-kb resolution matrices. Enrichment values are relative to a random background.

albumin. MNase (Thermo, #EN0181) concentrations were chosen to yield approximately 80% to 90% of mono-nucleosomal, according to a pretitrated digestion tests. Next, nuclei were resuspended in 200 µL of MBuffer (10 mM Tris-HCl, pH 8.0, 50 mM NaCl, 5 mM $MgCl_2$, 1 mM $CaCl_2$, 0.1% Triton X-100, and 1× Roche complete EDTA-free). Chromatin was fragmented using the appropriate amount of MNase for 10 min at 37 °C. The digestion was terminated by the addition of ethylene glycol-bis(β-aminoethyl ether)-N,N,N',N'-tetraacetic acid (EGTA; 1.5 mM final concentration) at 65 °C for 10 min. DNA 3' ends were dephosphorylated using 5 units of recombinant shrimp alkaline phosphatase (New England Biolabs, #M0203) at 37 °C for 45 min. The reaction was terminated by incubation at 65 °C for 5 min. Next, DNA 5' ends were phosphorylated using 20 units of T4 polynucleotide kinase (New England Biolabs, #M0201L) at 37 °C for 15 min. DNA overhangs were filled using 40 units of DNA Polymerase I, Large Klenow Fragment (New England Biolabs, #M0210L) with biotin-dCTP and biotin-dATP (Invitrogen) at 25 °C for 45 min. The reaction was terminated by incubation at 65 °C for 20 min with EDTA (0.03 M final concentration). Chromatin was collected, washed in 1× ligase buffer, and ligated using 50 units of T4 DNA Ligase (New England Biolabs, #M0202L) at room temperature for 3 h. After proximity ligation, biotin from unligated ends was removed using 200 units of exonuclease III (New England Biolabs, #M0206S) at 37 °C for 5 min. After cross-links had been reversed by incubation with 250 µg of proteinase K at 65 °C for 2 h, the ligated DNA was extracted using the Qiagen DNeasy Plant Mini Kit (Qiagen, #69106), in accordance with the manufacturer's instructions. DNA was purified and 250−400-bp fragments were selected and purified using 0.7× + 0.3× Ampure XP beads, which were then subjected to blunt-end repair, polyadenylation, and adaptor addition using the VAHTS Universal Plus DNA Library Prep Kit for MGI (Vazyme, #NDM617). The fragments were then subjected to streptavidin C1 beads (Invitrogen, #65001)-mediated pull-down and polymerase chain reaction (PCR) amplification (95 °C for 3 min; 11–14 cycles of 98 °C for 20 s, 60 °C for 15 s, and 72 °C for 30 s; 72 °C for 5 min; 4 °C hold). Amplified products were purified with 0.8× AMPure XP beads. Finally, Micro-C-XL libraries were quantitated and sequenced on the MGI-Seq platform (BGI, China).

**ChIP-Seq library construction and sequencing**
Briefly, 1–1.5 g aerial parts of seedlings were fixed in 1% formaldehyde for 15 min, and then 100 mmol/L glycine was added to terminate the fixation reaction. Chromatin was extracted with Nuclei Isolation Buffer (NIB) and sheared using a Bioruptor® Pico (Diagenode) (NIB: 50 mmol/L HEPES/Tris (NaOH) (pH = 7.4), 5 mmol/L $MgCl_2$, 25 mmol/L NaCl, 5% sucrose, 30% glycerin, 0.25% Triton™ X-100, 0.1% β- mercaptoethanol (β-ME), 0.1% SIGMA protease inhibitor). The sample was centrifuged

and the supernatant was incubated with an RNA Pol II CTD antibody (mAb) (Active Motif, 39097, 2 ng) or Goat anti-mouse IgG (H + L) antibody (HRP-conjugated) (EASYBIO, BE0102-100, 1:200 dilute) overnight at 4 °C. Immunocomplexes were captured with RPROTEIN A SEPHAROSE FF (Cytiva, 17127901) for 2 h at 4 °C. The beads were washed five times to remove nonspecific binding, and immunocomplexes were then eluted. After reversal of crosslinking overnight at 65 °C, DNA was extracted with phenol:chloroform:isoamyl alcohol (25:24:1) and precipitated with ethanol. Libraries were constructed using the VAHTS® Universal DNA Library Prep Kit for Illumina V3 (Vazyme, ND607) according to the manufacturer's instructions. DNA was amplified by 13 cycles of PCR during library construction. Libraries were sequenced on the NovaS4-150 PE system to generate 2× 150-bp paired-end reads. For each experiment, two biological replicates were performed.

## RNA-Seq for *Arabidopsis*, rice, and soybean
The fresh plant tissues were frozen in liquid nitrogen and stored at −80 °C. Tissues were ground into a fine powder using a mortar and pestle in liquid nitrogen. Total RNA was extracted using the Quick RNA Isolation Kit (Huayueyang, 0416-50gk) and treated with DNase I to remove DNA contamination. Libraries were constructed using the VAHTS® Universal RNA-Seq Library Prep Kit for Illumina (Vazyme, NR605-01) according to the manufacturer's instructions and sequenced on the NovaS4-150 PE system.

## RNA-Seq bioinformatics analysis
Low-quality reads and adapter sequences were removed using fastp[61], and the clean reads were mapped to the genome using STAR[62] (the genome versions used for different high-throughput sequencing analyses were consistent). The read counts table was generated by featureCounts using code from https://github.com/Linhua-Sun/Ath_Heat_Hi-C[17]. The expression levels of genes were compared between treatment and wild-type groups using strict criteria (FDR < 0.05 and fold change > 2) using DESeq2[63].

## Hi−C bioinformatics analysis
Hi−C contact matrixes for *Arabidopsis*, rice, and soybean were generated by the HiC-Pro pipeline in accordance with our published method[17]. Briefly, raw reads were filtered by fastp[61] to remove adapters and low-quality reads. The remaining data were subjected to the HiC-Pro pipeline[64], which included mapping, pair filtering, and ligation validity control. Bowtie2[65] was used to align both members of each pair to the reference genomes of *Arabidopsis* (TAIR10, http://www.arabidopsis.org/), rice (IRGSP1.0, http://rice.uga.edu/), and soybean (Wm82.a2.v1, https://www.soybase.org/). A two-step approach in HiC-Pro was used to rescue chimeric read pairs that included read-jumping junction sites. Low-quality, singleton, and multiple-mapped reads were discarded. The remaining well-mapped read pairs were assigned to fragments according to reference genomes and restriction enzyme cutting sites. Next, pairs from different restriction fragments were selected to build Hi−C contact maps and PCR duplicates were removed. Contact matrices were transformed into mcool files with multiple resolutions (e.g., 200-bp, 400-bp, 800-bp, 1,000-bp, 2,000-bp, 5,000-bp, 10,000-bp, 20,000-bp, 40,000, and 100,000-bp) using coolers from valid contact pairs generated by the HiC-Pro pipeline. Cooler balance was used to normalize Hi−C contact matrices.

## Micro-C-XL bioinformatics analysis
Micro-C-XL contact matrices were generated using the Dovetail Genomics pipeline (https://micro-c.readthedocs.io). Briefly, similar to Hi−C analysis, adapters and low-quality reads were removed by fastp[61], and read pairs were mapped to the reference genomes of three species using BWA-MEM (https://github.com/lh3/bwa). Then, PCR duplicates and reads with low mapping quality (<40) were discarded by pairtools

(https://github.com/open2c/pairtools). *Arabidopsis* mcool files that included multiple fixed bin sizes (e.g., 200-bp, 400-bp, 800-bp, 1000-bp, 2000-bp, 4000-bp, 5000-bp, 10,000-bp, 20,000-bp, 30,000-bp, 40,000-bp, 50,000-bp, and 100,000-bp bins) were generated by cooler (https://github.com/open2c/cooler). Soybean and rice mcool files also included larger bin sizes (200 kb and 500 kb). Cooler balance was used to generate normalized contact matrices in mcool files. Pairing comparisons of Hi−C, and Micro-C-XL were conducted using CoolBox[66]. CoolBox was also used to calculate the matrix of observed/expected and draw heat maps. Distance-dependent decay of chromatin contact density plots was generated using a modified version of hicPlotDistVsCounts[17]. ATAs for PCGs, domains, and boundaries were generated using coolpup.py[67]. Insulation score analysis was conducted to call boundaries, with the top 25% of locally lowest regions regarded as significant borders, using different bin sizes established by cooltools. Boundary distributions were assessed using ChIPseeker (https://bioconductor.org/packages/release/bioc/html/ChIPseeker.html). Metagene plots and heat maps of insulation scores were generated by deeptools[68].

The association analyses of tRNA genes and chromatin boundaries were performed using regioneR (https://bioconductor.org/packages/release/bioc/html/regioneR.html).

Generalized linear regression analysis was used to calculate the top predictive factors for boundary strength. Subsequently, the weighted mean values from various ChIP-Seq/ATAC-Seq were calculated for every 200-bp region in the genome by bedmap[69]. A matrix was then constructed together with the insulation scores and analyzed using the generalized linear model (GLM) in R. Large-scale comparisons of Micro-C maps among samples were mainly adopted from our Hi−C methods[17]. Relative difference heatmaps at 100-kb resolution were generated by HiCDatR[70] and visualized using our previous Hi−C visualization method[17]. To quantitatively demonstrate differences in chromatin organization between WT and mutant or treatment Micro-C-XL at the 100-kb resolution, IDEs were calculated with HiCExplorer[71] using a modified version of hicPlotDistVsCounts[17]. Chromatin contact over the gene body was calculated by an average z-score method adopted from previous research using code from (https://github.com/heard-lab/HiCExplorer/tree/SummarizePerRegion/hicexplorer). The weighted average of Pol II CTD chromatin occupancy over the gene body was also calculated by bedmap.

To effectively correlate the ChIP-Seq data for various proteins at the 1D level with the 3D data provided by Micro-C-XL, we used 200-bp resolution data and the peak information provided by ChIP-Seq to search for any possible interactions between two peaks within the 50-kb range (all possible sub-matrixes are included). We extracted these interactions and compared them with a random background, finally calculating an average to construct an APA map with which to determine whether each protein factor has the ability to participate in the chromatin loop at the genome-wide level (also see Fig. 7a). This method was achieved using coolpup.py[67].

## ChIP-Seq and ATAC-Seq bioinformatics analysis
Publicly available ChIP-Seq/ATAC-Seq datasets[20,28,42,48,72–85] were used in this study, detailed information of which is provided in Supplementary Data 3. We selected these public datasets based on their similarity to plant tissue and the growth period used. Fastp was used to remove low-quality reads and adapters from raw sequencing reads[61]. The remaining clean reads were aligned to the corresponding reference genome by Bowtie2[65] (v1.2.1.1), as indicated in the Hi−C analysis subsection, with two mismatches. PCR duplicates were identified by Picard's MarkDuplicates (http://broadinstitute.github.io/picard/). Only uniquely mapped reads were subjected to analysis via SAMtools[86]. bamCoverage in deepTools[68] was used to generate bigwig files with options that included binSize 10, normalizeUsingRPKM, and effectiveGenomeSize for different species. *Arabidopsis* blacklist regions[72]

were discarded using the option --blackListFileName in bamCoverage in deeptools[68]. Typical ChIP-Seq loci were plotted using Integrative Genomics Viewer software[87]. Some ChIP-Seq bigwig files were downloaded from ChIP-Hub (https://biobigdata.nju.edu.cn/ChIPHub/). Metagene plots were generated by SeqPlots (https://github.com/Przemol/seqplots). Shuffled SE regions were generated by bedtools shuffle[88].

## Post hoc analyses of chromatin boundaries and SEs

Insulation analyses were performed using the insulation module (with different bins) in cooltools. An integrated heatmap with many genomic and epigenomic annotations was generated for each locus containing an SE. Each SE was subjected to a detailed manual assessment of chromatin organizing patterns and then classified into four categories (loop, stripe, domain, and other). Mate gene plots of transcription factors, chromatin-associated factors, and other ChIP-Seq datasets were generated and then used to identify factors possibly involved in SE regulation (analysis by SeqPlots). Global identification of possible TFs within chromatin borders is based on previously published research[89].

## Statistics and reproducibility

No statistical method was used to predetermine the sample size. No data were excluded from the analyses. The experiments were not randomized. The investigators were not blinded to allocation during experiments and outcome assessment.

## Reporting summary

Further information on research design is available in the Nature Portfolio Reporting Summary linked to this article.

## Data availability

The Micro-C-XL, RNA-Seq, and ChIP-Seq data generated in this study have been deposited in the National Genomics Data Center Genome Sequence Archive database under accession code PRJCA014302. The *Arabidopsis* chromatin boundaries data generated in this study are provided in the Source Data file. Source data are provided in this paper. The *Arabidopsis* leaf DNase-Seq bigwig file was downloaded from PlantDHS. *Arabidopsis* SEs were downloaded from a recently published article[29]. Enhancers from *Arabidopsis* leaves were extracted from a published study[28]. Chromatin state information was collected from published data[34]. Source data are provided in this paper.

## Code availability

The custom code used for the analysis has been deposited in GitHub.

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

## Acknowledgements

This study was supported by the Key Program of National Natural Science Foundation of China (32230006 to X.W.D.), the Director's Award of Peking University Institute of Advanced Agricultural Sciences (to H.H.), the Shandong Development Fund of Science & Technology, the Key R&D Program of Shandong Province (ZR202211070163 to L.S.), and the Award of Natural Science Foundation of Shandong Province (ZR2021ZD30).

Bioinformatics analyses were performed at the High-Performance Computing Facility of Peking University Institute of Advanced Agricultural Sciences.

## Author contributions

H.H. and L.S. conceived and designed the project. L.S. performed all bioinformatics analyses. L.S., J.Z., X.X., W.N., L.Z., and Y.T.L. performed the experiments and managed sequencing. L.S. constructed the figures and tables with input from J.Z., Y.L., and N.M. L.S. drafted the paper. H.H. and X.W.D. revised the paper and supervised the work.

## Competing interests

The authors declare no competing interests.
