## [Peer Review File · Nature Communications]

Mapping nucleosome-resolution chromatin organization and enhancer-promoter loops in plants using Micro-C-XLREVIEWER COMMENTS

Reviewer #1 (Remarks to the Author):

With the prevailing approach named Hi-C, genome-wide chromatin organization patterns in plants have been intensively investigated over the last decade. However, because Hi-C maps rely on the distribution of restriction enzyme cutting sites, it is impossible to comprehensively examine chromatin organization at a sub-kilobase resolution. This limit, in particular, hinders plant scientists who wish to better understand fine-scale chromatin interactions in the model plant species *Arabidopsis*, as it has a crowded and compact genome. In this study, the authors attempted to explore nucleosome-scale (200bp) chromatin interaction networks of *Arabidopsis* by using Micro-C-XL.

Firstly, the authors demonstrated that this method, which Hsieh and colleagues initially developed, could outperform the traditional Hi-C method in terms of revealing local scale chromatin interactions (fig1). Next, the authors demonstrated that at a local scale, the *Arabidopsis* genome organization showed clear domains and insulated regions, and their location is closely linked to multiple factors, among which gene transcription (Pol2), gene layout (convergent, divergent, or tandem), and epigenetic marks were highlighted. Subsequently, the authors demonstrated that the Micro-C-XL data could reveal the interactions between previously identified super-enhancers and their target genes. Lastly, the authors presented the implication of Micro-C-XL on crop species.

In my opinion, this manuscript marks an important step in unveiling local chromatin organizations in plants. The manuscript was well written; figures were clearly annotated. On the other hand, this manuscript generated limited insights, as most of its conclusions, for example, the correlation between chromatin domain boundaries and Pol2, have already been drawn from several previous Hi-C studies. With such high-resolution micro-C-XL datasets and the many available publically available datasets, perhaps the authors can try to establish a model, with which local chromatin interaction patterns (e.g., the location and the strength of insulation) can be formulated. Similar approaches have been done for metazoan HiC maps, and I was wondering if it is the right time to do so in plants.

Another point: I noticed that very often Hi-C and Micro-C-XL maps have blank strips in which no reads mapping occurred. I understand that these regions might be full of repeats (i.e., they are solely mapping problems). Besides, some blank areas on Hi-C maps may also be due to the absence of RE cutting sites. Nevertheless, I am curious about the appearance of "Micro-C-XL-specific" blank strips, for example, in Fig5B, FigS5E, and several other figures. What are the technical/biological reasons behind them?

Minor comments:

It was written in line 117 that the Micro-C-XL was modified. It is better to explicitly describe what modifications have been applied in the Methods part.

I could not find "Figure 3F" (mentioned in line 255) in the manuscript.

It is unclear with which criteria the authors selected public ChIP-seq datasets and what ChIP-seq datasets have been re-analyzed. The part between lines 334 and 350 is not well connected to the rest part of this manuscript.

The two replicates of Pol-III ChIP-seq (Fig 3B) appear to have a large variation. As an alternative, the authors can consider checking the association between domain boundaries and tRNA loci.

The section title on page 9 (“...without epigenetic modification”) is misleading. Perhaps the correlating epigenetic mark has not been tested. For the chromatin interaction highlighted in Fig4C, how about trying to associate them with an ATAC-seq track?

I don't think that Fig4A, which is a snapshot, well supports the claim at the beginning of page 11 (“Simultaneous analysis of ... mainly occurred...”). Statistical tests are also required.

If gene expression turns out to be a strong correlation factor with local chromatin organization in Arabidopsis, I wonder why expression data was omitted from Fig6. For Fig6B, why was the CHG track specifically shown?

Reviewer #2 (Remarks to the Author):

Sun et al. has used Micro-C-XL to map the chromatin interactions in Arabidopsis. My biggest concern is that this paper has no biology in it. It is just a big data dump without generating any new insight. What they described in the result (domain border, transcription, chromatin stats etc) are well known facts. Compare to Hi-C, their new Micro-C data indeed has much better resolution, and they have successfully demonstrated that. But could they use this tool/technology to solve some actual plant biology question? Not those hypothetical ones. If not, no one will be interested in reading it. Since the method is well known. Applying it to plant tissue doesn't deserve a publication.

I think it is a rather dangerous trend here. People saw a lot of sequencing paper in top tier journals. But they often forgot that a method paper only has value when it is relatively new. Once the method is developed, we should use them to solve real questions. Not just repeat the method in another species and claim that we have re-discovered some well known principles in genomics. Maybe it is better to let the journal's editorial side to give a clear guideline for future authors. Otherwise, the whole research field will be wasting funding and manpower to do such meaningless experiments.

Reviewer #3 (Remarks to the Author):

General Comments

Sun and colleagues present a very nice data set in their manuscript. The clarity of the obtained contact matrices is impressive. They also performed additional analyses allowing to better correlate 1D and 3D genome organization.

On a more critical side, I was a bit confused, as the manuscript swings between methods and research paper, yet not really being one of the two. The discussion section does not include enough comments about the limitations and specific advantages of micro-C. For example, given one is interested in larger resolution, should one also preferentially employ micro-C. What are potential biases leading to misinterpretation of micro-C data

(sequence composition, nucleosome density..). In contrast, the authors at large discuss proteins that may facilitate 3D contacts. Proteins, which were only introduced in a supplementary figure. Given the presented data, I am not 100% convinced that these proteins are truly involved in forming contacts (although they mostly have been considered as candidates before). Given the length of this discussion, it is indeed a bit surprising that the authors did not test them. It seems to me that there was not too much experimental work needed for the entire manuscript, thus this additional experiment might have been worthwhile.

In short, the data is impressive, yet the type of manuscript is unclear. I would suggest that during a reviewing process, the manuscript should get a more pronounced focus on either technical or biological aspects.

Specific Comments:

P1 line 25.. Please rephrase, as RNA Pol is not a boundary (this is the way I understand the phrase)

P3 line 63-65: It would be nice to include a reference here or explain the basis of this claim. At large scale, 3D architecture varies considerably among species (e.g. chromosome configurations)

P5 line 140: Why would double crosslinking lead to more cis-contact capture?

P6 line 145: Why is it cost-effective? I couldn't find any information on the read length, yet assuming 100nt reads, the authors acquired 2G reads of which 400M are informative. Yielding 1/5 informative contacts seems to me rather standard and, thus, as "cost-effective" as other Hi-C approaches.

P6 line 148: By "unique" contacts, the authors refer to contacts that could not be found in the other approaches?

P6 line 168.: Indeed, restriction enzyme sites are unevenly distributed. However, nucleosome density is also not uniform along chromosomes. How did the authors deal with the effect of general chromatin compaction density (i.e. nucleosome density) on their analysis? Additionally, MNase has a nucleotide bias as certain genomic regions do (e.g. enhancers). How does this affect the analysis?

Figure 1E: The authors state that canonical Hi-C is not suitable to analyze contacts below 1kb resolution. Whereas I am impressed by the "clarity" of the Micro-C-XL contact matrix, I think it is somewhat "unfair" to compare the two at 200bp resolution. Four-cutters cut in average every 256nt, thus it is anticipated that 200bp bin size does not produce a meaningful result for canonical Hi-C. How does the comparison look like at 1kb resolution?

Figure 2A: The clear distinction of folding domains in the shown genomic region is indeed impressive. It would be nice to also include a chromatin accessibility track, which would give a better indication whether nucleosome density may represent a source to influence the shown results.

Figure 2F and 2G: I am somewhat confused about CS4 in 2F. It seems to me that CS4 is depleted in boundaries. How comes that the authors claim that it contributes to boundary

formation (line 206...)? Furthermore, what are boundaries with “greatest strength”. Also the different chromatin states do not occur in equal numbers, for example only very few regions are covered by CS9. How is this imbalance taken into account. Maybe the authors could explain their findings here a bit better?

Figure 2H. In the legend there is a typo as it says (F) instead of (H). Also, what do the axes describe? On the x-axis: the fraction of peaks of a given TF that overlaps with boundaries? The y-axis: The absolute number of overlaps? This is not very informative, as I assume that the absolute number of peaks varies considerably between different TFs? Thus, would it make more sense to just use the x-axis data represented as a bar plot?

P8 line 220: Maybe this phrase should include a reference?

Figure 3D: I cannot really see a difference in the orange gradient between the plots. Maybe there could be a better way to represent the classes in transcriptional activity.

Figure 3E: The silenced genes seem to make less clear boundaries as the authors suggest. However, they seem to make more 3'-5' loops than highly expressed genes. Could it be that highly expressed genes occur more in TAD-like structures (crumples), whereas the weakly expressed ones form loops? I am aware that the paper they cite states otherwise. Maybe the authors could comment on this? Small remark: In the text the authors mention Fig3F, which does not exist (line 251).

P9, line 258: The results do not show that transcription has an effect on chromatin architecture but only that the two somewhat correlate.

P19, line 265: In my opinion H3K27me3 is not a heterochromatic mark.

Figure 4C. Which new class of novel interactions do the authors mean. An arrow or more details explaining it in the main text would be appreciated.

P11, line 301: There should be a reference here.

P11, line 308: “we identified strong correlation..” What does that mean? Is there a p-value or a summary figure that could support the claim?

P12, line 327: The text refers to 5D, yet 5E is meant.

Supplementary Figure 6: The metaplots all show an enrichment for the respective proteins towards -500 and +500 bp distant to the SE. How can the authors explain this. Maybe the interpretation would rather be that the respective proteins are depleted in the SE borders? The authors should cite the origin of the ChIP-seq data they used for their analysis.

P12, line 343..: I am not so sure about the model. Here the authors depict (if I interpret the cartoon correctly) that multiple sites along the enhancer contacts a specific regions of the promoter (based on the triangles). However, in 5B this looks rather the opposite, where the entire ERF017 contacts the enhancer.

P13. Line 378: What do the authors mean by two libraries? I would have assumed that the number of sequenced reads is important. I am not aware of the term library in terms of the amount of information content.

P15, line 420: I guess this is true for Hi-C but not for microscopy based observation....

P17, line 483: "Chromatin was fragmented using the appropriate amount of MNase for 10 min at 37°C". Well, it would be of great interest what the appropriate amount of MNase is!

P19, line 531: Why did the authors use a different aligner for Hi-C and micro-C?

P19: How did the authors calculate the observed/expected contacts. I could not find this information in the methods section.

We are grateful for the helpful feedback provided by the reviewers. In response to their concerns, we have added more data and revised our manuscript. We hope that the revisions are sufficient and the manuscript is now acceptable for publication. Please find our detailed responses to the reviewers' comments listed below.

REVIEWER COMMENTS

Reviewer #1 (Remarks to the Author):

With the prevailing approach named Hi-C, genome-wide chromatin organization patterns in plants have been intensively investigated over the last decade. However, because Hi-C maps rely on the distribution of restriction enzyme cutting sites, it is impossible to comprehensively examine chromatin organization at a sub-kilobase resolution. This limit, in particular, hinders plant scientists who wish to better understand fine-scale chromatin interactions in the model plant species *Arabidopsis*, as it has a crowded and compact genome. In this study, the authors attempted to explore nucleosome-scale (200bp) chromatin interaction networks of *Arabidopsis* by using Micro-C-XL.

Firstly, the authors demonstrated that this method, which Hsieh and colleagues initially developed, could outperform the traditional Hi-C method in terms of revealing local scale chromatin interactions (fig1). Next, the authors demonstrated that at a local scale, the *Arabidopsis* genome organization showed clear domains and insulated regions, and their location is closely linked to multiple factors, among which gene transcription (Pol2), gene layout (convergent, divergent, or tandem), and epigenetic marks were highlighted. Subsequently, the authors demonstrated that the Micro-C-XL data could reveal the interactions between previously identified super-enhancers and their target genes. Lastly, the authors presented the implication of Micro-C-XL on crop species.

1. In my opinion, this manuscript marks an important step in unveiling local chromatin organizations in plants. The manuscript was well written; figures were clearly annotated. On the other hand, this manuscript generated limited insights, as most of its conclusions, for example, the correlation between chromatin domain boundaries and Pol2, have already been drawn from several previous Hi-C studies. With such high-resolution micro-C-XL datasets and

the many available publically available datasets, perhaps the authors can try to establish a model, with which local chromatin interaction patterns (e.g., the location and the strength of insulation) can be formulated. Similar approaches have been done for metazoan HiC maps, and I was wondering if it is the right time to do so in plants.

Response: Thank you for the positive comments.

a) In response to the questions raised by Reviewer 1 and the comments made by the other two reviewers, we conducted a thorough study on the manner by which RNA Pol II affects the structure of chromatin. Our goal was to take the relationship between RNA Pol II and chromatin domains from correlation to causality. Designing specific experimental systems is necessary to validate the role of Pol II in regulating chromatin structure. Unlike yeast cells and mammalian cell lines (for example, the auxin-inducible degron (AID) system is widely used in the field of 3D genomics, which can rapidly degrade targeted proteins and detect the effects on chromatin structure), knockdown of Pol II in plant systems is relatively complex. For rigorous analysis, we designed two methods: 1) using a special weak allele *nrb2-3*, which can obtain a certain amount of material for high-throughput sequencing experiments; and 2) using the exogenously applied inhibitor, FVP, which specifically affects transcriptional elongation and impedes the function of Pol II. Both these strategies can interfere with Pol II while still maintaining plant survival. Based on the genetic and chemical interference strategies, we carried out a series of experiments (Micro-C-XL, Pol II CTD ChIP-seq, and RNA-seq) and related in-depth analysis to demonstrate that Pol II can regulate local chromatin organization.

Relevant experimental methods, schemes, and results have been integrated into the main text, Fig. 4, and Supplementary Figs. 6 and 7. All generated data have been uploaded to the public database. We also hope that our strategy can be used to study chromatin structure in plants.

b) Based on the suggestions, we have also established a model to illustrate the top factors related to the strength and location of boundaries. We focused on the use of a generalized linear model (GLM) by converting ChIP-seq data for different factors into values per 200 bp, integrating with the strength of the chromatin boundary (expressed by the insulation score),

and further calculating the correlation coefficient of each factor using the GLM. As shown in Supplementary Fig. 4, we used the existing public data series, which we had used in the first version of our manuscript, and CHIP-seq datasets for the Pol V, MORC7, and MED12 proteins. We found that ATAC-seq can best predict the strength of chromatin boundaries in *Arabidopsis*. This information is indeed missing from the first version of the manuscript, and Reviewers 1 and 3 also suggest integrating ATAC-seq into the figure of typical loci. The results of our model and the information on typical loci are consistent, and ATAC-seq is a very good predictive indicator. The chromatin boundary strength prediction data have been incorporated into the main text.

For the analysis of the location of chromatin boundaries, the main difference is that the information regarding each 200-bp position is encoded into the information of 0/1 (using chromHMM). The presence or absence of boundaries is also defined as the 0/1 vector, after which GLM is used for analysis. We obtained the 0/1 matrix by contacting the corresponding authors of two recently published works ^{1,2}. The results indicate that H3K4me3 is a relatively strong predictor (Fig. R1). However, as can be seen directly from Fig. 2a, the peaks of H3K4me3 and ATAC-seq are very close, but there is still a slight difference in the distance between their peaks and chromatin boundaries. If only calculated in the form of 0/1 at 200-bp resolution, it may cause bias. Considering that CHIP-seq is used to encode the form of 0/1 per 200 bp and relies only on chromHMM, we do not feel it is accurate enough to predict the location of chromatin boundaries.

Therefore, we have integrated the prediction of chromatin boundary strength into the main text. The calculation of strength is linear, and there is no simple dichotomy; thus, we believe that the accuracy is higher. Mammalian models still use 1 kb of information for analysis, but 1-kb resolution is insignificant for *Arabidopsis*.

Note: Considering that the research on chromatin boundaries and loops in *Arabidopsis* has just begun; for example, mammalian CTCF-like factors have not yet been identified, we need to combine Micro-C-XL/ChIP-seq or ChIA-PET technology to mine key factors and analyze their potential biological functions. A complete and more accurate model of chromatin structure in plants will be established in the future.

Fig. R1. | Heatmap of the regression coefficients from the generalized linear model. Prediction of chromatin boundary location using binarized matrixes of multiple ChIP-seq datasets from two recently published studies. **a** Multiple histone modifications/variants were used ¹. **b** Multiple histone modifications were used ².

- Another point: I noticed that very often Hi-C and Micro-C-XL maps have blank strips in which no reads mapping occurred. I understand that these regions might be full of repeats (i.e., they are solely mapping problems). Besides, some blank areas on Hi-C maps may also be due to the absence of RE cutting sites. Nevertheless, I am curious about the appearance of “Micro-C-XL-specific” blank strips, for example, in Fig5B, FigS5E, and several other figures. What are the technical/biological reasons behind them?

Response: Thank you for raising this very meaningful question. We have conducted the following analysis to address this issue. If we needed to analyze the specific blank regions of Micro-C-XL, we first extracted the respective blank regions of Hi-C and Micro-C at a resolution of 200 bp using cooltools. As shown in Fig. R2a, the gray area under the heatmap body represents the extracted blank regions. Firstly, we observed that the overlap between Hi-C and Micro-C-XL blank regions is not high in the overlay analysis. Due to the significant differences in enzyme digestion and sequencing, Hi-C has more discrete blank regions (Fig. R2a, b). Comparing the length distribution of the two, it was found that Hi-C has longer and shorter blank regions than Micro-C-XL (Fig. R2c). Further analysis of the specific blank regions of Micro-C-XL in comparison with the corresponding random regions revealed that overall, the raw reads coverage is much lower, which indicates problems with the alignment

of these regions (Fig. R2d). By showing a locus, we can also observe that Micro-C-XL-specific peaks are located in the transposon region, and the coverage of data alignment is very low (Fig. R2d, e). We further analyzed these Micro-C-XL-specific blank regions as well as the known transposons and pericentromeres in *Arabidopsis* and found that most Micro-C-XL-specific blank regions overlap with transposons and/or pericentromeres (Fig. R2f).

In summary, the specific blank regions of Micro-C-XL mainly arise from regions with mapping issues and low coverage such as TE and pericentromeres.

Fig. R2 | Micro-C-XL-specific blank regions. **a** Typical loci with called Hi-C and Micro-C-XL blank regions. **b** Overlap between Hi-C and Micro-C-XL blank regions. **c** Length distribution of Hi-C and Micro-C-CL blank regions. **d** Micro-C-XL raw reads coverage distribution for Micro-C-XL-specific blank regions and the corresponding shuffled regions. **e** Typical IGV screenshot showing the raw read coverage and gene/TE for annotations of those blank regions. **f** Overlap analysis among Micro-C-XL-specific blank regions, TE annotations, and pericentromeric regions.

Minor comments:

3. It was written in line 117 that the Micro-C-XL was modified. It is better to explicitly describe what modifications have been applied in the Methods part.

Response: Three minor modifications were made. Firstly, we adjusted the composition of the MB buffer to make it suitable for plants (replacing Tris HCl pH = 7.5 with Tris HCl pH = 8.0 and 0.2% NP-40 with 0.1% Triton X-100). Secondly, after reversal of crosslinking, we used the Qiagen DNeasy Plant Mini Kit to extract DNA (instead of phenol:chloroform:isoamyl alcohol extraction). Thirdly, we used 0.7X + 0.3X Ampure XP beads to select and purify DNA of the target fragment length (250–400 bp) (instead of agarose gel extraction). The description of experimental modifications has been updated in the revised manuscript.

4. I could not find “Figure 3F” (mentioned in line 255) in the manuscript.

Response: We apologize for this error; we have revised it.

5. It is unclear with which criteria the authors selected public ChIP-seq datasets and what ChIP-seq datasets have been re-analyzed. The part between lines 334 and 350 is not well connected to the rest part of this manuscript.

Response: Our strategy was to select data as close to the type of our plant tissues and stages as possible. Information regarding the ChIP-seq datasets that we used has been summarized in Supplementary Data 3.

The second question is related to a major comment made by Reviewer 3. Accordingly, we have revised the language and added new experiments to further strengthen the novel insights that Micro-C-XL can provide (chromatin loops and possible protein factors). The previous analysis only had one-dimensional information of ChIP-seq and lacked association with 3D genomic datasets. We have optimized this as much as possible in the new version of the manuscript. Please see new Fig. 7 and associated main text.

6. The two replicates of Pol-III ChIP-seq (Fig 3B) appear to have a large variation. As an alternative, the authors can consider checking the association between domain boundaries and tRNA loci.

Response: Thank you for the suggestion. This can effectively and concisely demonstrate the relationship between the two, avoiding the drawback of the tDNA length being too small to analyze. We found that there is indeed a significant overlap between the positions of tRNA genes and chromatin boundaries (Fig. R3a). This overlap was also demonstrated to be significant ($p < 0.001$) via a permutation test using the R package regioneR (<https://bioconductor.org/packages/release/bioc/html/regioneR.html>) (Fig. R3b). We have made slight modifications to the main text.

Fig. R3 | Overlap and statistical analysis between tRNA genes (a) and chromatin boundaries (b).

7. The section title on page 9 (“...without epigenetic modification”) is misleading. Perhaps the correlating epigenetic mark has not been tested. For the chromatin interaction highlighted in Fig4C, how about trying to associate them with an ATAC-seq track?

Response: We have made the corresponding modifications. According to the prompt, information such as ATAC-seq has been added to the figure. Of course, the main purpose is to introduce the following text pertaining to chromatin loops and super-enhancers (Fig. 6 and Fig. 7); therefore, we intentionally did not include ATAC-seq information in the previous version of the manuscript.

8. I don't think that Fig4A, which is a snapshot, well supports the claim at the beginning of page 11 ("Simultaneous analysis of ... mainly occurred..."). Statistical tests are also required.

Response: Thank you for raising this statistical issue. Indeed, this phenomenon is difficult to analyze statistically, so we have changed the description here. We will explain from other perspectives; please see new Fig. 7 and the associated main text.

9. If gene expression turns out to be a strong correlation factor with local chromatin organization in Arabidopsis, I wonder why expression data was omitted from Fig6. For Fig6B, why was the CHG track specifically shown?

Response: Thank you for your comments. We have performed RNA-seq experiments in rice and soybean and added the results to the figure (from initial Fig. 6a, b to updated Fig. 8a, b). Both our own work and a series of existing studies have found a strong correlation between the heterochromatin H3K9me2 and local chromatin structure³⁻⁵. We generally use H3K9me2 modification as a reference for the local chromatin organization map. Moreover, CHG DNA methylation and H3K9me2 are not only highly correlated in terms of genomic distribution but also due to the cyclic information loop between the writers of the two in terms of biochemical structure in plants^{6,7}. Thus, for species like soybean that lack H3K9me2 modification ChIP-seq, we opted for CHG DNA methylation from WGBS.

Reviewer #2 (Remarks to the Author):

10. Sun et al. has used Micro-C-XL to map the chromatin interactions in Arabidopsis. My biggest concern is that this paper has no biology in it. It is just a big data dump without generating any new insight. What they described in the result (domain border, transcription, chromatin stats etc) are well known facts. Compare to Hi-C, their new Micro-C data indeed has much better resolution, and they have successfully demonstrated that. But could they use this tool/technology to solve some actual plant biology question? Not those hypothetical ones. If not, no one will be interested in reading it. Since the method is well known. Applying it to plant tissue doesn't deserve a publication.

I think it is a rather dangerous trend here. People saw a lot of sequencing paper in top tier journals. But they often forgot that a method paper only has value when it is relatively new. Once the method is developed, we should use them to solve real questions. Not just repeat the method in another species and claim that we have re-discovered some well known principles in genomics. Maybe it is better to let the journal's editorial side to give a clear guideline for future authors. Otherwise, the whole research field will be wasting funding and manpower to do such meaningless experiments.

Response: Thank you for the evaluation of our work. Firstly, it is not easy to apply a new technology to plants. We have long focused on the field of plant 3D genomics, but traditional technologies such as Hi-C, 3C, and FISH also seriously limit our research on high-resolution chromatin structures in plants, which is why we have put a lot of effort into Micro-C-XL. After 2020, high-quality research in mammals and fruit flies rapidly emerged using/developing Micro-C-related tools⁸⁻¹⁸. At the same time, we believe that our work is not only the successful establishment of Micro-C-XL technology in plants but also its application in the discovery of a series of biological insights and new research directions.

Firstly, we can accurately identify clean chromatin domains based on the resolution advantage of Micro-C-XL over Hi-C. We have added a series of new experiments to validate that Pol II can regulate local chromatin folding.

Another key result of Micro-C-XL is the widespread presence of visible chromatin loops in the *Arabidopsis* 3D genome (Fig. 6 and Fig. 7), which was difficult to accurately identify using Hi-C technology in the past. We have conducted more in-depth research on these issues from both analytical and experimental perspectives. The raw interaction map shows the correlation between protein binding sites and short-range chromatin loops (< 50 kb). This is significant for studying the local chromatin structure and gene expression regulation in *Arabidopsis*. The greatest significance of our work is further delving into nucleosome resolution in plants, laying the foundation for future detailed research on transcriptional regulation. High-resolution technologies can enhance our confidence in the field of plant 3D genomics.

We have added experiments and analysis to enhance the correlations to causal relationships. We hope that our work can be reconsidered.

Our highlights are:

- a) Nucleosome-resolution chromatin conformation capture was first established in plants.
- b) The association between gene transcription and local higher-order chromatin structure was found, and Pol II interference experiments validated that Pol II can regulate gene-level chromatin structure.
- c) The presence of chromatin loops/strips closely related to enhancers was found in *Arabidopsis*, indicating that it can be applied to mine various *cis*-regulatory elements (enhancer/silencer) involved in chromatin loops. We identified a series of protein factors that may be involved in the regulation and formation of chromatin loops.
- d) Further, Micro-C-XL was also tested in other crops, such as rice and soybean, demonstrating the advantages of this technology in plant higher-order chromatin structure analysis.

Reviewer #3 (Remarks to the Author):

General Comments

11. Sun and colleagues present a very nice data set in their manuscript. The clarity of the obtained contact matrices is impressive. They also performed additional analyses allowing to better correlate 1D and 3D genome organization.

On a more critical side, I was a bit confused, as the manuscript swings between methods and research paper, yet not really being one of the two. The discussion section does not include enough comments about the limitations and specific advantages of micro-C. For example, given one is interested in larger resolution, should one also preferentially employ micro-C. What are potential biases leading to misinterpretation of micro-C data (sequence composition, nucleosome density..). In contrast, the authors at large discuss proteins that may facilitate 3D contacts. Proteins, which were only introduced in a supplementary figure. Given the presented data, I am not 100% convinced that these proteins are truly involved in forming contacts (although they mostly have been considered as candidates before). Given the length of this

discussion, it is indeed a bit surprising that the authors did not test them. It seems to me that there was not too much experimental work needed for the entire manuscript, thus this additional experiment might have been worthwhile.

In short, the data is impressive, yet the type of manuscript is unclear. I would suggest that during a reviewing process, the manuscript should get a more pronounced focus on either technical or biological aspects.

Response: Thank you for the positive comments and suggestions.

a) The question regarding the article type is important. Our work demonstrates the first time that nucleosome resolution has been achieved in plants, which is of great significance from the perspective of methodology; however, our paper should be a research article and we have further strengthened our research data accordingly. We have added two other key points: (1) research on Pol II knockdown to verify the regulation of chromatin structure by Pol II and (2) analysis and experiments related to specific protein-mediated chromatin loops. This makes our paper a typical research article and significantly improves the conclusion from relevance to causality. We have provided sufficient experimental verification, new analytical ideas, and more biological insights. Potential biases have also been analyzed and discussed (Question 17).

b) Although this is a research article, our results/methods for Micro-C-XL also deserve more introduction to the field. With respect to the advantages and disadvantages of Micro-C-XL, we have also made a few revisions to the discussion. Regarding the questions raised, we believe that Micro-C-XL can also be used for large-scale analysis, reflected in Supplementary Fig. 2, Supplementary Fig. 7, and Supplementary Fig. 11. Micro-C-XL can bear the large-scale of Hi-C. It is not cost-effective to only perform large-scale analysis with Micro-C-XL when considering economic benefits.

c) We mainly answer the question regarding specific proteins and chromatin loops from two perspectives. Firstly, both Reviewer 1 and 3 raised questions about the analysis of typical loci and potential regulators associated with super-enhancers, and mentioned that some statistical analysis is needed; however, difficulties exist at present. Therefore, we established a new pipeline to effectively correlate these two types of information from 1D ChIP-seq and 3D Micro-C-XL, with a view to quantitating whether a protein has the potential to participate in the formation and/or maintenance of chromatin loops and found a new factor (Pol V). Two

examples clearly show the close relationship between MORC/Pol V and specific chromatin loops.

Secondly, from an experimental perspective, we also strongly believe that identifying regulators of chromatin structure in plants is the central issue of plant 3D genomics at present, which is essential for the subsequent study of mechanisms and functions. This may also be the reason why our emphasis was a little unbalanced in the first version of the manuscript. Accordingly, we further analyzed these three types of protein factors in depth (Fig. 7). We have conducted in-depth research on the chromatin loops and MORCs/PolV and updated the relevant results in Fig. 7 and Supplementary Fig. 11 of this version. We also further clarified the potential of MORCs to mediate short-range chromatin loops, which has not been described in previous studies. Of course, from a genetics perspective, there are six active genes in the MORC protein family; therefore, we did not directly test mutants such as *morc6*, but instead performed a series of experiments and analyses using RdDM mutants (based on the fact that RdDM recruits MORC7). Although it is regrettable that we did not obtain positive results, we have provided new insights. In the future, direct experiments in the *morc* hexuple (*morchex*, in which all functional MORCs are knocked out) mutant should be performed.

According to the results shown in the new version of the manuscript (Fig. 7), MED12 may indeed be involved, but its functions are relatively weak. We also attempted to conduct relevant research; however, *med12* has a strong phenotype and it is difficult to obtain enough seeds for various high-throughput sequencing experiments in a short period of time. Therefore, we chose *cdk8-1*. In the CDK8/MED12/MED13 module, the phenotype of *cdk8-1* is relatively weak with respect to growth under normal conditions, and it is easy to obtain enough seeds. We performed Micro-C-XL and other experiments in *cdk8-1*; unfortunately, CDK8 ChIP-seq using a GFP transgenic line was not successful after multiple attempts. To avoid confusion in the field, we have not updated these results in the article.

It is worth noting that, we can observe from our analysis results that the interaction patterns mediated by these different types of protein factors are different. Moreover, our observation of a series of typical loci also found considerable heterogeneity in the short-range chromatin loops in *Arabidopsis*, indicating that further in-depth research is needed in the future. Similar to SYD-related loops, we hypothesize that they may play a role in changing chromatin

accessibility, not necessarily a direct motor protein, which is not the focus of our study.

In summary, we have conducted a series of experiments and more in-depth analysis, hoping to answer the concerns and questions.

Specific Comments:

12. P1 line 25.. Please rephrase, as RNA Pol is not a boundary (this is the way I understand the phrase)

Response: Thank you for the advice. We have revised this sentence.

13. P3 line 63-65: It would be nice to include a reference here or explain the basis of this claim. At large scale, 3D architecture varies considerably among species (e.g. chromosome configurations)

Response: Thank you for your advice. We have included a reference to support our statement. We fully agree with this viewpoint and would like to emphasize that while large-scale structures like chromosome territories and AB compactions exist in both animals and plants, small-scale results such as domains and loops are highly variable and frequently under debate.

14. P5 line 140: Why would double crosslinking lead to more cis-contact capture?

Response: Thank you for your question. Randomly ligated, un-crosslinked fragments may diffuse freely, resulting in increased *trans* contacts. Double crosslinking can reduce random ligation events, which leads to an increase in *cis* contacts and a decrease in *trans* contacts¹⁹.

15. P6 line 145: Why is it cost-effective? I couldn't find any information on the read length, yet assuming 100nt reads, the authors acquired 2G reads of which 400M are informative. Yielding 1/5 informative contacts seems to me rather standard and, thus, as "cost-effective" as other Hi-C approaches.

Response: Sorry for the misunderstanding. Emphasizing this point serves no merit; therefore, we have removed it to avoid confusion.

16. P6 line 148: By “unique” contacts, the authors refer to contacts that could not be found in the other approaches?

Response: Sorry for the misunderstanding; we have removed it to avoid confusion.

17. P6 line 168.: Indeed, restriction enzyme sites are unevenly distributed. However, nucleosome density is also not uniform along chromosomes. How did the authors deal with the effect of general chromatin compaction density (i.e. nucleosome density) on their analysis? Additionally, MNase has a nucleotide bias as certain genomic regions do (e.g. enhancers). How does this affect the analysis?

Response: Thank you for these very important questions. Micro-C-related research can easily raise concerns about whether these issues exist. We have consulted a series of literature and carried out further analyses based on our own data with the hope of addressing these concerns.

a) Nucleosome density and normalization issue.

There is essentially no difference between the traditional Hi-C and Micro-C-XL. Hi-C also has problems such as GC bias, unequal sequencing depth, and enzyme biases. Similar to Hi-C, Micro-C solves this problem using the classic matrix-balancing algorithm (implicit method to correct systemic biases) “Knight-Ruiz (KR)” or “Iterative Correction and Eigenvector decomposition (ICE)”^{20,21}. All noise sources are handled “implicitly.” These two similar algorithms are essentially based on the assumption that, in theory, every bin in the genome should have equal rights, that is, the so-called same visibility; therefore, a process similar to coverage normalization can be continuously carried out. Until the end, each bin gets equal visibility or so-called convergence. This method has been widely used in the analysis of Hi-C and Micro-C data.

b) MNase bias issue.

Regarding MNase- and Micro-C-related bias, especially issues related to chromatin accessibility, Tsung Han S. Hsieh and colleagues provided a very detailed explanation in a public review document of a recently published *Nature Genetics* article (<https://static->

8/MediaObjects/41588_2022_1223_MOESM3_ESM.pdf, page 36-38). We have referred to this information in the revised text and also performed further analysis of our own data.

The laboratories of Steven Henikoff and Gernot Längst carried out a series of experiments and analyses demonstrating that MNase can target different regions without preference (Fig. R4), including those with different accessibility (defined by DNase-seq), euchromatin or heterochromatin, and loci with different transcriptional levels. Steven Henikoff reached the conclusions following an in-depth analysis of MNase-seq (Chereji, R.V. *et al.*).

Fig. R4 | MNase can assess different regions equally. This figure was directly copied from Chereji *et al.* in Fig. 9. The rightmost panel includes green (HP1-bound), yellow (active), red (active), blue (Polycomb-bound), and black (repressive) chromatin states defined in (Filion *et al.*, 2010). All figures are from Chereji, R.V., Bryson, T.D., and Henikoff, S. (2019). Quantitative MNase-seq accurately maps nucleosome occupancy levels. *Genome Biology*.

Based on the series of articles, we concluded that MNase can uniformly cleave chromatin regardless of accessibility.

c) Micro-C bias issue.

It is both interesting and noteworthy that Micro-C can also be considered an MNase-seq with ultra-deep sequencing depth^{15,22} therefore, the analysis bias of the two technologies should be similar. In addition, several articles have analyzed the bias issue of Micro-C and its relationship with DNase/ATAC-seq open chromatin regions^{9,11,13,14,23}. There is no evidence of bias toward

open chromatin regions in Micro-C.

Generally, it is not believed that there is a bias toward enhancers using Hi-C. The situation in *Arabidopsis* is very different from that in mammals; for instance, there is a lack of classic examples of in-depth study of EP loops. Based on some examples identified by Micro-C-XL, we can also find the corresponding interaction in Hi-C at different resolutions but the interaction region in Hi-C is not clear (Fig. R5a, b). This further demonstrates why Micro-C-XL is needed to lay a good foundation for future plant mechanism and function research.

Next, we treated the Micro-C-XL data as conventional ChIP-seq data, ignoring the paired information. At this time, we calculated the original reads coverage per 1-kb bin, as well as calculating the signal values per 1-kb bin for ATAC-seq. No correlation was found between the two. Replacement of the Micro-C part with the original summed contact value per 1-kb bin still yielded no correlation (Fig. R5c). Moreover, we divided the peaks of ATAC-seq into three categories and calculated the distribution of the two separately. ATAC-seq showed more open regions, while the original reads coverage depth of Micro-C was lower. In conclusion, we can exclude the influence of MNase preference, especially on open chromatin or enhancers (Fig. R5d).

In plants, MNase cleavage can be applied to obtain information on chromatin accessibility (called MH-seq), which can be used to identify enhancers²⁴. It is worth noting that the working concentration of MNase used in this work was relatively low, which may be the reason why MNase is thought to have a preference for enhancers.

Fig. R5 | Micro-C-XL quality control for the chromatin accessibility issue. a, b Two typical loci (containing potential super-enhancers) are shown. The differences in chromatin structure between Hi-C and Micro-C-XL techniques are shown at different resolutions (from 200 bp to 800 bp). **c** The peaks of ATAC-seq were divided into three categories based on signal intensity, and the distribution of ATAC-seq signals and the coverage of Micro-C raw sequencing reads were observed. **d** The signal strength of every 1-kb region of ATAC-seq was compared with the original reads coverage of Micro-C-XL and the original unnormalized contact counts. No correlation was found between the ATAC-seq and Micro-C-XL from either analysis.

18. Figure 1E: The authors state that canonical Hi-C is not suitable to analyze contacts below 1kb resolution. Whereas I am impressed by the “clarity” of the Micro-C-XL contact matrix, I think it is somewhat “unfair” to compare the two at 200bp resolution. Four-cutters cut in average every 256nt, thus it is anticipated that 200bp bin size does not produce a meaningful result for canonical Hi-C. How does the comparison look like at 1kb resolution?

Response: This is indeed a good question. Firstly, there is currently no better comparison strategy, since there are no other types of data suitable for comparison with Micro-C-XL. Hi-C is still the gold standard in the entire 3D genome field. Secondly, although Hi-C can theoretically cut the genome better, the distribution of restriction enzyme sites is relatively uneven and does not simply cut every 256 nt. Thirdly, comparison of 200 bp (Micro-C-XL) vs. 200 bp (Hi-C) is generally used between Micro-C (or other similar approaches) and Hi-C in mammals ^{9,11,14,23}.

Finally, we also generated heatmaps at different bin sizes such as 200-bp, 400-bp, 800-bp, and 1000-bp (Fig. R6). From any graph, the advantages of Micro-C-XL in analyzing local chromatin organizations and chromatin loop interactions in comparison with Hi-C can clearly be observed. In general, Hi-C may also be able to observe certain chromatin structures (~ 1 kb or < 1 kb, usually rough and blurry), but Micro-C-XL makes these structures very clear; and of course, new structures can also be discovered. We believe that this is of great significance for future in-depth research on the mechanisms and functions of 3D plant structures.

Fig. R6 | Direct comparisons between Hi-C and Micro-C-XL at different resolutions.

Two typical loci are shown to compare the differences in chromatin structure between the Hi-C and Micro-C-XL techniques at different resolutions (from 200-bp to 1000-bp).

19. Figure 2A: The clear distinction of folding domains in the shown genomic region is indeed impressive. It would be nice to also include a chromatin accessibility track, which would give a better indication whether nucleosome density may represent a source to influence the shown results.

Response: According to the suggestion, we have added ATAC-seq and DNase-seq to the corresponding main and supplementary figures, which is very helpful. We have specifically described the results and GLM prediction in the main text.

20. Figure 2F and 2G: I am somewhat confused about CS4 in 2F. It seems to me that CS4 is depleted in boundaries. How comes that the authors claim that it contributes to boundary formation (line 206...)? Furthermore, what are boundaries with "greatest strength". Also the different chromatin states do not occur in equal numbers, for example only very few regions are covered by CS9. How is this imbalance taken into account. Maybe the authors could explain their findings here a bit better?

Response: Thank you for the suggestions and questions. Our description of CS4 here was incorrect and we have removed it. Boundaries with the "greatest strength" were determined according to the absolute value for the insulation score of the identified chromatin boundaries. The greater the absolute value for the insulation score, the greater the intensity. This is consistent with the observations in typical loci (Fig. 2a and Supplementary Fig. 3).

We used the enrichment score to account for the varying number of CS regions. To be more precise, it is better to use the ratio of the number of boundaries that overlap with CS to the total number of specific CS types. The absolute value and the actual number of overlaps are also marked on the bars. Referring to the first question asked by Reviewer 1, at the end of the paragraph, we also updated the GLM model analysis and found that ATAC-seq is a good predictor of boundary strength, which we ignored in the initial manuscript.

21. Figure 2H. In the legend there is a typo as it says (F) instead of (H). Also, what do the axes describe? On the x-axis: the fraction of peaks of a given TF that overlaps with boundaries? The y-axis: The absolute number of overlaps? This is not very informative, as I assume that the

absolute number of peaks varies considerably between different TFs? Thus, would it make more sense to just use the x-axis data represented as a bar plot?

Response: We apologize for this error; it has been revised. Thank you for raising this issue. Indeed, it does not make sense to use the numbers on the y-axis. We mainly considered two points. Firstly, the absolute number of TF binding is indeed influenced by many factors, such as the quality of antibodies or IP and the TF protein itself. However, the number of peaks for certain TFs, such as CTCF and YY1, is also a very important factor and has been widely studied in mammals. There is an obvious feature that their binding peaks are widely distributed in the genome. Such TFs may have a higher probability of participating in the global control of chromatin structure. Another simple consideration is to mark the names of some TFs to separate these scattered points.

22. P8 line 220: Maybe this phrase should include a reference?

Response: Thank you for the suggestion. Our previous version of the manuscript cited three classic works; therefore, we have added another piece of recent literature posted on bioRxiv.

23. Figure 3D: I cannot really see a difference in the orange gradient between the plots. Maybe there could be a better way to represent the classes in transcriptional activity.

Response: Thank you for the advice; this has been changed.

24. Figure 3E: The silenced genes seem to make less clear boundaries as the authors suggest. However, they seem to make more 3'-5' loops than highly expressed genes. Could it be that highly expressed genes occur more in TAD-like structures (crumples), whereas the weakly expressed ones form loops? I am aware that the paper they cite states otherwise. Maybe the authors could comment on this? Small remark: In the text the authors mention Fig3F, which does not exist (line 251).

Response: We apologize for this error; it has been corrected. The issue regarding gene expression level and 3'-5' loops formed on the gene body proposed by the reviewer has been

relatively vague in the field for a long time. We have conducted a systematic analysis of this issue, as described below.

The question of whether there is a chromatin loop at the gene level comes from research in yeast²⁵. As early as 2012, a gene loop was identified in yeast, and a key protein SSU72 was identified, suggesting a model that SSU72 can mediate gene loops²⁵. Subsequently, in 2015, using the early version of Micro-C, the concept of crumples was proposed (similar to gene folding)²⁶. In 2016, a series of gene loops were identified using Hi-C technology in *Arabidopsis* (this definition is different from the gene loops we defined here and does not require the formation of typical dot-like interactions/loops between TSS and TTS), which is more similar to the concept of crumples proposed by yeast Micro-C²⁷. Subsequently, Tsung Han S. Hsieh and his colleagues found only a small number of gene loops under strict conditions when analyzing mouse Micro-C data in 2020¹⁴. Gene loops are not ubiquitous cases in mammalian cells¹⁴ (Fig. R7).

We also reanalyzed the yeast Pol II ChIP-seq and Micro-C data^{26,28}. Indeed, as Pol II binding becomes stronger in yeast, chromatin interaction weakens, showing a negative correlation (Fig. R8a–c). However, this weakening is not a simple change in the intensity of internal interaction within the structure but a change in the interaction patterns, which also differs from *Arabidopsis*. At the same time, we can observe that there is indeed a possibility of gene looping over genes with a high binding intensity of Pol II in yeast (Fig. R8b, c).

Firstly, it is necessary to define what state is which structure. Based on the latest research in mammals, we believe it can be divided into two types of structures. When the entire gene body region is covered by high-strength interactions, which we call gene folding. When a dot-like structure is formed between the head (TSS) and tail (TTS) of a gene, and the background is very clear, we call it gene looping (Fig. R8d).

Secondly, gene length should be considered. Recent work by other groups has also found that gene length plays an important role in the analysis of chromatin structure over the gene body²⁹. Therefore, we also classified chromatin structure at the gene level in *Arabidopsis* based

on gene length and expression. The results are shown in the figure, which are similar to those in mammals and are greatly affected by length (Fig. R9).

In summary, as a comment, there are indeed differences among different species; however, in mammals and *Arabidopsis*, the phenomenon of gene loops is rare and not universal. In *Arabidopsis*, highly expressed and longer genes have a high interaction intensity, which we call gene folding (equivalent to crumples). This is consistent with the conclusion made by Prof. Liu Chang in 2016; only the statements and analysis methods are different. As for genes with lower expression, it can be seen from our ATA maps that genes with longer lengths have little trend toward gene looping. Short genes seem possible, but dot-like interactions/loops are not obvious (Fig. R9).

Thus, we do not believe that genes with lower expression in *Arabidopsis* will form so-called gene loops, or 3'-5' loops. Of course, we cannot rule out the possibility of a few individual cases. We have indeed observed a few examples, such as TE genes, where both the head and tail are co-bound to Pol V and MORC, which may form weak dot-like loops; however, this is, to some extent, beyond the scope of this study.

Fig. R7 | Gene loop identified by mouse Micro-C. This figure is directly copied from Supplemental Figure 5i in Hsieh, T.-H.S. *et al.*, (2020). Resolving the 3D landscape of transcription-linked mammalian chromatin folding. *Molecular Cell* 78, 539-553.e8. Mouse Micro-C identified 303 significant gene loops. Their model showing gene loops and Pol II and SSU72 has been provided.

Fig. R8 | Pol II and gene-level chromatin organization in yeast. **a** Pol II ChIP-seq metagene over genes from lower Pol II binding (Q1) to higher Pol II binding (Q9). **b, c** ATA over each type of gene using EGS-based (**b**) and DSG-based (**c**) Micro-C. **d** Schematic diagram showing gene looping and folding.

Fig. R9 | Chromatin organization patterns considering gene length and gene expression level. As the direction of the arrow increases horizontally, the length of the gene gradually increases. Vertically, with the direction of the arrow, the gene expression gradually increases.

25. P9, line 258: The results do not show that transcription has an effect on chromatin architecture but only that the two somewhat correlate.

Response: Thank you for the comment. This is a very important issue. Based on the suggestions from all three reviewers, we carefully designed a Pol II knockdown experiment, which fully demonstrated that Pol II can regulate local chromatin structure. The relevant response can be found by referring to the first comment by Reviewer 1. At the same time, we have added many figures (Fig. 4 and Supplementary Figs. 6 and 7) to illustrate this. Moreover, the relevant text, methods, etc., have been updated.

26. P19, line 265: In my opinion H3K27me3 is not a heterochromatic mark.

Response: We have modified this description.

27. Figure 4C. Which new class of novel interactions do the authors mean. An arrow or more details explaining it in the main text would be appreciated.

Response: Thank you for the advice; we have added an arrow.

28. P11, line 301: There should be a reference here.

Response: Thank you for the advice; we have added a reference.

29. P11, line 308: “we identified strong correlation..” What does that mean? Is there a p-value or a summary figure that could support the claim?

Response: This question is similar to question 8, which we have already answered. It is indeed difficult to calculate the statistical situation from this perspective; therefore, we have changed our approach, mainly referring to Fig. 7.

30. P12, line 327: The text refers to 5D, yet 5E is meant.

Response: We apologize for this error; it has been revised.

31. Supplementary Figure 6: The metaplots all show an enrichment for the respective proteins towards -500 and +500 bp distant to the SE. How can the authors explain this. Maybe the interpretation would rather be that the respective proteins are depleted in the SE borders? The authors should cite the origin of the ChIP-seq data they used for their analysis.

Response: Thank you for the question, which is worthy of consideration. We cannot rule out phenomena such as those protein factors being depleted at the boundary of SE. Therefore, we have changed the description in the revised manuscript.

It is worth noting that the definition of *Arabidopsis* super-enhancer is determined based on the clustered intergenic open chromatin regions found by ATAC-seq/DNase-seq. The definition of borders of super-enhancers is relatively rough in the screenshot of IGV (see Supplementary Figs. 9 and 10). It is also difficult to perform statistical analysis at the genome-wide level, as mentioned in the response to the statistical issues raised (questions 8 and 29).

Meanwhile, screenshots based solely on IGV cannot provide a good understanding of the protein factors associated with chromatin interactions related to enhancers or super-enhancers. We have conducted in-depth research on this issue in the newly added main Fig. 7. We have attempted to establish an effective solution that combines the peaks of ChIP-seq with interaction data for analysis in a relatively unbiased manner.

We have summarized the information of all public ChIP-seq data used and placed it in Supplementary Data 3. Citations have been made in the main text.

32. P12, line 343..: I am not so sure about the model. Here the authors depict (if I interpret the cartoon correctly) that multiple sites along the enhancer contacts a specific regions of the promoter (based on the triangles). However, in 5B this looks rather the opposite, where the entire ERF017 contacts the enhancer.

Response: We apologize for the misunderstanding. This kind of super-enhancer generally has multiple peaks with open chromatin, as defined in a previous study; therefore, multiple triangle annotations are used. At the same time, these super-enhancers may interact with the promoters

and/or gene body of target genes, etc. Of course, in *Arabidopsis*, chromatin loops and related structures have high heterogeneity. The current model is simplified. We fine-tuned this model, especially the loop on the left side, to avoid misunderstanding that there is an interaction between multiple sites.

33. P13. Line 378: What do the authors mean by two libraries? I would have assumed that the number of sequenced reads is important. I am not aware of the term library in terms of the amount of information content.

Response: Here, the library refers to a pool of DNA fragments from the Micro-C library construction experiment containing adapters compatible with a specific sequencing platform. Generally, the product is obtained from an independent biological replicate.

The reviewer's opinion is correct, and there is no doubt that sequencing depth significantly impacts the quality of Hi-C/Micro-C-XL data. This is another issue worthy of consideration. Indeed, the deeper the sequencing depth, the better the data and map obtained. However, it should be noted that the library complexity for Micro-C-XL or Hi-C is limited, and increasing depth will not always have a gaining effect (conventional 1D ChIP-seq and RNA-seq are not so sensitive to sequencing depth). As the amount of data gradually increases, the read duplicates in the library will also increase but the number of unique contacts will only increase very slowly, gradually reaching a plateau.

Therefore, this is a balancing act. Simply increasing the sequencing of the library generated by one experiment cannot achieve a high-resolution map. Multiple experiments need to be performed, each sequencing to a certain depth (in our experience, sequencing to 200–300 Gb is generally suitable), to achieve good results by pooling two or more libraries. For example, the Micro-C map of mouse ECS cells achieved nucleosome resolution by merging 38 libraries

14.

We have modified this part of the discussion slightly.

34. P15, line 420: I guess this is true for Hi-C but not for microscopy based observation....

Response: We apologize that our statement was not rigorous enough; it has been revised in the new version of the manuscript.

35. P17, line 483: “Chromatin was fragmented using the appropriate amount of MNase for 10 min at 37°C”. Well, it would be of great interest what the appropriate amount of MNase is!

Response: Chromatin was fragmented using the appropriate amount of MNase for 10 min at 37°C according to the titration results. For example, we divided the crosslinked nuclei into three equal portions, then took one and divided that into five equal smaller portions. Different MNase amounts (such as 2U, 4U, 6U, 8U, 10U) were added for digestion, and DNA was extracted for agarose gel electrophoresis. If 6U is appropriate (~80% mononucleosomes), we take another portion of the crosslinked nuclei with 30U MNase for formal digestion. Each sample needs to be titrated first. Due to the incomplete quantity of crosslinked nuclei from different samples, the amount of MNase titrated varies; therefore, the amount of MNase used for formal digestion is also different. This has been described in published Micro-C articles^{14,28}; thus, we did not elaborate here.

36. P19, line 531: Why did the authors use a different aligner for Hi-C and micro-C?

Response: In fact, it is very simple. We and our collaborators have used Hi-C to study the chromatin structure in plants³⁰⁻³². All our previous studies employed HiC-Pro, a widely used Hi-C data pre-processing pipeline. Our current Hi-C results should be 100% consistent with our previously published data. Micro-C data sets are usually very large and generally need to be analyzed at high resolution. In comparison with Hi-C analysis, our Micro-C analysis was mainly conducted at a scale of less than 1000 bp. Many traditional analysis processes are no longer workable or have low efficiency. Considering the calculation speed, intermediate file storage, final data visualization, and analyses, we mainly refer to some new processes and tools from 4DN. For example, bwa mem-based pipelines are more CPU and memory-friendly. The core is to obtain useful contact information.

37. P19: How did the authors calculate the observed/expected contacts. I could not find this information in the methods section.

Response: We apologize for not describing this clearly. We used the expect function in CoolBox to calculate the matrix of observed/expected contacts and to construct the heatmaps (https://gangcaolab.github.io/CoolBox/_gallery/reso_normalize.html). We have added this information to the methods (Micro-C-XL bioinformatics analysis).

References:

1. Jamge, B. *et al.* Histone variants shape chromatin states in *Arabidopsis*. *eLife* **12**, RP87714 (2023).
2. Fu, L.-Y. *et al.* ChIP-Hub provides an integrative platform for exploring plant regulome. *Nat. Commun.* **13**, 3413 (2022).
3. Feng, S. *et al.* Genome-wide Hi-C analyses in wild-type and mutants reveal high-resolution chromatin interactions in *Arabidopsis*. *Mol. Cell* **55**, 694–707 (2014).
4. Zhao, L. *et al.* Chromatin loops associated with active genes and heterochromatin shape rice genome architecture for transcriptional regulation. *Nat. Commun.* **10**, 1–13 (2019).
5. Peng, Y. *et al.* Chromatin interaction maps reveal genetic regulation for quantitative traits in maize. *Nat. Commun.* **10**, 2632 (2019).
6. Law, J. A. & Jacobsen, S. E. Establishing, maintaining and modifying DNA methylation patterns in plants and animals. *Nat. Rev. Genet.* **11**, 204–220 (2010).
7. Du, J., Johnson, L. M., Jacobsen, S. E. & Patel, D. J. DNA methylation pathways and their crosstalk with histone methylation. *Nat. Rev. Mol. Cell Biol.* **16**, 519–532 (2015).
8. Li, X. *et al.* GAGA-associated factor fosters loop formation in the *Drosophila* genome. *Mol. Cell* **83**, 1519-1526.e4 (2023).
9. Goel, V. Y., Huseyin, M. K. & Hansen, A. S. Region Capture Micro-C reveals coalescence of enhancers and promoters into nested microcompartments. *Nat. Genet.* **55**, 1048–1056 (2023).
10. Barshad, G. *et al.* RNA polymerase II dynamics shape enhancer–promoter interactions. *Nat. Genet.* **55**, 1370–1380 (2023).
11. Hsieh, T.-H. S. *et al.* Enhancer–promoter interactions and transcription are largely maintained upon acute loss of CTCF, cohesin, WAPL or YY1. *Nat. Genet.* **54**, 1919–1932 (2022).
12. Swygert, S. G. *et al.* Local chromatin fiber folding represses transcription and loop extrusion in quiescent cells. *eLife* **10**, e72062 (2021).

13. Hua, P. *et al.* Defining genome architecture at base-pair resolution. *Nature* **595**, 125–129 (2021).
14. Hsieh, T.-H. S. *et al.* Resolving the 3D landscape of transcription-linked mammalian chromatin folding. *Mol. Cell* **78**, 539–553.e8 (2020).
15. Krietenstein, N. *et al.* Ultrastructural details of mammalian chromosome architecture. *Mol. Cell* **78**, 554–565.e7 (2020).
16. Mohana, G. *et al.* Chromosome-level organization of the regulatory genome in the *Drosophila* nervous system. *Cell* **0**, (2023).
17. Levo, M. *et al.* Transcriptional coupling of distant regulatory genes in living embryos. *Nature* **605**, 754–760 (2022).
18. Batut, P. J. *et al.* Genome organization controls transcriptional dynamics during development. *Science* **375**, 566–570 (2022).
19. Akgol Oksuz, B. *et al.* Systematic evaluation of chromosome conformation capture assays. *Nat. Methods* **18**, 1046–1055 (2021).
20. Imakaev, M. *et al.* Iterative correction of Hi-C data reveals hallmarks of chromosome organization. *Nat. Methods* **9**, 999–1003 (2012).
21. Rao, S. S. P. *et al.* A 3D map of the human genome at kilobase resolution reveals principles of chromatin looping. *Cell* **159**, 1665–1680 (2014).
22. Huang, Y., Wang, B. & Liu, J. NucleoMap: A computational tool for identifying nucleosomes in ultra-high resolution contact maps. *PLoS Comput. Biol.* **18**, e1010265 (2022).
23. Krietenstein, N. *et al.* Ultrastructural details of mammalian chromosome architecture. *Mol. Cell* **78**, 554–565.e7 (2020).
24. Zhao, H. *et al.* Genome-wide MNase hypersensitivity assay unveils distinct classes of open chromatin associated with H3K27me3 and DNA methylation in *Arabidopsis thaliana*. *Genome Biol.* **21**, 24 (2020).
25. Tan-Wong, S. M. *et al.* Gene loops enhance transcriptional directionality. *Science* **338**, 671–675 (2012).
26. Hsieh, T.-H. S. *et al.* Mapping nucleosome resolution chromosome folding in yeast by Micro-C. *Cell* **162**, 108–119 (2015).
27. Liu, C. *et al.* Genome-wide analysis of chromatin packing in *Arabidopsis thaliana* at single-

gene resolution. *Genome Res.* **26**, 1057–1068 (2016).

28. Hsieh, T.-H. S., Fudenberg, G., Goloborodko, A. & Rando, O. J. Micro-C XL: assaying chromosome conformation from the nucleosome to the entire genome. *Nat. Methods* **13**, 1009–1011 (2016).

29. Salari, H., Fourel, G. & Jost, D. Transcription regulates the spatio-temporal dynamics of genes through micro-compartmentalization. *BioRxiv* (2023) doi:10.1101/2023.07.18.549489v1.

30. Sun, L. *et al.* Conserved H3K27me3-associated chromatin looping mediates physical interactions of gene clusters in plants. *J. Integr. Plant Biol.* **65**, 1966–1982 (2023).

31. Yang, T. *et al.* Chromatin remodeling complexes regulate genome architecture in *Arabidopsis*. *Plant Cell* **34**, 2638–2651 (2022).

32. Sun, L. *et al.* Heat stress-induced transposon activation correlates with 3D chromatin organization rearrangement in *Arabidopsis*. *Nat. Commun.* **11**, 1886 (2020).

REVIEWERS' COMMENTS

Reviewer #1 (Remarks to the Author):

After careful checking, I conclude that the authors have appropriately addressed my concerns, and revised their manuscript accordingly. Thanks!

Reviewer #2 (Remarks to the Author):

Great, the authors now depleted the pol2 and found that chromatin organization has been changed. So isn't all these so call 3D chromatin structure just a result of transcription? It seems I have reached a point where I no longer feel compelled to contribute further insight or critique for something has no value. Nevertheless, I hold the hope that in the coming years, scientists will stumble upon these seemingly purposeless papers on plant chromatin organization and, upon encountering these comments, find amusement in their frivolity.

Reviewer #3 (Remarks to the Author):

I would like to thank the authors for all their efforts to improve the manuscript. They answered most of my comments and substantial new data was added. Unfortunately, their new analysis did not add any "ground breaking" new insights into Arabidopsis 3D genome architecture. I would like to stress that the data and analysis thereof looks very nice, yet the link between transcription and 3D genome folding has been already proposed many years ago. It is rather surprising how small the observed effect of the mutant is, which is a very interesting and valuable result per se.

The authors also improved the discussion, especially, concerning the pros and cons of Micro-C XL, which I highly appreciate.

Overall, the authors here provide a very nice and usefull data set and the added value of Micro-C XL to study 3D genome folding is clearly shown. Unfortunately, the paper cannot provide completely new biological insights but rather confirms what has been known or suspected before. I assume that the authors are fully aware of that, as the discussion starts with technical aspects and also the conclusion mainly covers this. Thus, although strong in that perspective, the paper is of rather technical nature.

One last comment remains for me: What is the biological meaning of a chromatin stripe? Since they were first described in 2015, I always had trouble to fully grasp what they represent in a real nucleus (and not just in Hi-C data...).

Thank you very much for the careful work of the three reviewers. Please find our detailed responses to the reviewers' comments listed below.

REVIEWERS' COMMENTS

Reviewer #1 (Remarks to the Author):

After careful checking, I conclude that the authors have appropriately addressed my concerns, and revised their manuscript accordingly. Thanks!

Response: Thank you very much to the reviewer for carefully reviewing our article and acknowledging it.

Reviewer #2 (Remarks to the Author):

Great, the authors now depleted the pol2 and found that chromatin organization has been changed. So isn't all these so call 3D chromatin structure just a result of transcription? It seems I have reached a point where I no longer feel compelled to contribute further insight or critique for something has no value. Nevertheless, I hold the hope that in the coming years, scientists will stumble upon these seemingly purposeless papers on plant chromatin organization and, upon encountering these comments, find amusement in their frivolity.

Response: Thank you very much for the reviewer's comments.

The relationship between transcription and chromatin structure has always been a hot topic in the field of three-dimensional genomics. In our current work, we have demonstrated that the chromatin structure over gene body is affected by transcriptional elongation. Of course, we also want to emphasize that there are still many other types of chromatin structures in plants, and further research is needed to resolve the causal relationship.

Reviewer #3 (Remarks to the Author):

I would like to thank the authors for all their efforts to improve the manuscript. They answered most of my comments and substantial new data was added.

Unfortunately, their new analysis did not add any "ground breaking" new insights into Arabidopsis 3D genome architecture. I would like to stress that the data and analysis thereof looks very nice, yet

the link between transcription and 3D genome folding has been already proposed many years ago. It is rather surprising how small the observed effect of the mutant is, which is a very interesting and valuable result per se.

The authors also improved the discussion, especially, concerning the pros and cons of Micro-C XL, which I highly appreciate.

Overall, the authors here provide a very nice and usefull data set and the added value of Micro-C XL to study 3D genome folding is clearly shown. Unfortunately, the paper cannot provide completely new biological insights but rather confirms what has been known or suspected before. I assume that the authors are fully aware of that, as the discussion starts with technical aspects and also the conclusion mainly covers this. Thus, although strong in that perspective, the paper is of rather technical nature.

One last comment remains for me: What is the biological meaning of a chromatin stripe? Since they were first described in 2015, I always had trouble to fully grasp what they represent in a real nucleus (and not just in Hi-C data...).

Response: Thank the reviewers for their recognition of our last round of revision.

We recognized that the correlation between transcription and chromatin structure in plants has been reported several years ago. However, there is no genetic evidence for the causal relationship between the two. This is the first time that we have proved the causal relationship between the two through a variety of means in plants.

As for why the effect of mutants is so small, this is mainly because the system of plants is different from that in mammals. It is not based on a cell line. And we can't use an auxin-inducible degron (AID) like system to quickly remove a protein complex. Therefore, we changed our methods, used mutant (*nprb2-3*) and applied chemical reagents *in vitro* to reduce the activity of Pol II. At the same time, we need to maintain the survival of plants, avoiding excessive concentrations that can cause plant death. Of course, our central purpose is to elaborate that the chromatin structure at the gene level will change when Pol II binding/elongation is weakened.

The current work is indeed biased towards the technical level, and we hope to better study the plant chromatin architectures and functions through Micro-C-XL combined with plant molecular genetics in the future.

Finally, we also think it is an interesting question about chromatin stripe. First of all, we suspect that chromatin stripe is due to the fact that Hi-C/Micro-C-XL and other technologies are based on a large number of cells rather than single-cell. Therefore, it is possible that the information in different cells is merged together, that is why it is difficult to imagine what the state of intracellular stripe is. Therefore, single-cell Hi-C or single-cell Micro-C-XL may provide clues in the future. Meanwhile, CRISPR-Cas9 system can be used to edit anchor sites of chromatin stripes and clarify the functional importance.